# Low frequency oscillations – neural correlates of stability and flexibility in cognition

Julia Ericson [1] ✉, Nieves Ruiz Ibáñez[1], Mikael Lundqvist [2] & Torkel Klingberg [1] ✉

Cognitive processing relies on the brain's ability to balance flexibility for encoding new information with stability for maintaining it. We examined these dynamics in three magnetoencephalography (MEG) datasets of visuospatial working memory (vsWM) tasks. Across all tasks, we identified four distinct networks in the theta and alpha bands, which were used to define functional states. Optimal transitioning rate between states was associated with better cognitive performance. Further, two of the states were linked to flexibility and stability, respectively: an encoding state dominated by a posterior theta and a maintenance state dominated by a dorsal alpha. We simulated the states in an in-silico model with biologically realistic cortical connectivity. The model, featuring spiking and oscillatory cortical layers interacting via phase-amplitude coupling, demonstrated how frequency and spatial region could modulate information flow. Our findings suggest a cognitive control mechanism, where selective transitions between large-scale networks optimize information flow, enabling both stable and flexible visual representations.

A fundamental question in cognitive neuroscience is how the brain balances stability with flexibility[1,2]. Even in basic tasks such as performing saccades, the brain needs to alternate between flexibly sampling new information and generating visual stability during eye movements[3]. In tasks involving visuospatial working memory (vsWM), controlling flexibility and stability is crucial: while stable representations are necessary to maintain information, flexibility is required to encode or update information. To achieve this balance, the brain may alternate between different functional states, allowing for state-specific responses to identical stimuli[4]. Identifying the mechanisms that generate these states are key to understanding the dynamic nature of cognition.

One possibility is that low-frequency, large-scale synchronization could explain the mechanism behind functional brain states[5,6]. In mice it has been shown that brain activity from a diverse set of behaviors, ranging from rest and external stimulus response to engagement in social behavior, can be explained by a few low dimensional, spatiotemporal activation patterns[7]. In humans, similar large-scale, low-frequency activities in the theta (4–8 Hz) and the alpha range (8–14 Hz) have been observed in a range of perceptual and cognitive processes from the control of eye-movements[8] to complex cognitive tasks[9].

Even though the role of these low-frequency networks is not fully understood, it has been proposed that they influence high-frequency spiking through phase-amplitude coupling[10]. For example, magneto- and electroencephalography (M/EEG) studies have shown that the phase of theta and alpha activity is often coupled with the amplitude of gamma activity[11–14]. Intracranial recordings have also demonstrated how low-frequency oscillations are linked to high-frequency gamma bursts[10,15–17]. Thus, low-frequency, synchronized networks might control the flow of information contained within higher frequencies[18–20].

Here, we aimed to study the role of low-frequency synchronization in cognitive brain states using two independent datasets of MEG-

[1]Department of Neuroscience, Karolinska Institutet, Stockholm, Sweden. [2]Department of Clinical Neuroscience, Karolinska Institutet, Stockholm, Sweden. ✉e-mail: julia.ericson@ki.se; torkel.klingberg@ki.se

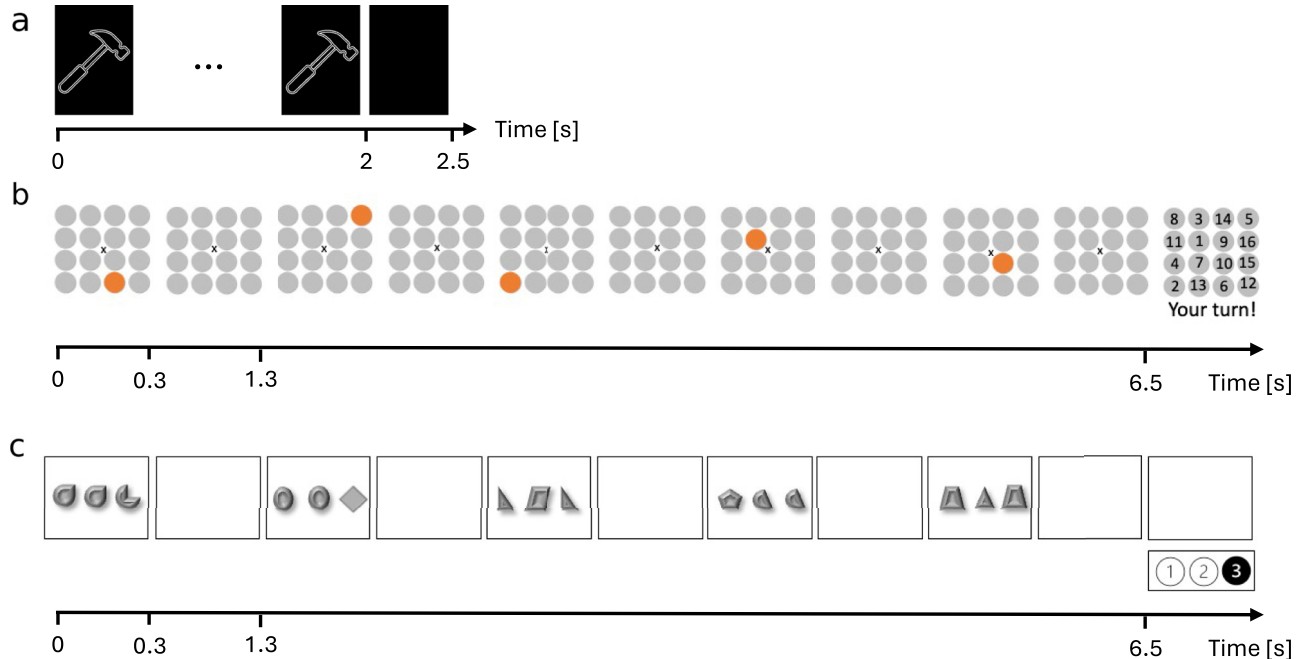

**Fig. 1 | The tasks from the HCP and the 4-subject dataset. a** The 2-back task from the HCP. Each new image is shown for two seconds. During this time, the subjects must report if the image matches the image from two trials earlier. This is followed by a 0.5 s memory delay. **b** The WM-Grid task from the 4-subject dataset. Each stimulus is displayed for 0.3 s, followed by a 1 s delay period. **c** The Odd One Out task from the 4-subject dataset. Just as for WM-grid, each stimulus is displayed for 0.3 s, followed by a 1 s delay period.

scans during vsWM tasks. From filtered MEG-data in the theta and alpha bands, we first defined synchronized networks using Independent Component Analysis (ICA), separately within each subject. We then clustered the network activities across subjects to find spatio-temporal activity patterns common to all subjects. These clusters were used to define brain states.

We first established a link between the control of the identified states and cognition by examining the relationship between state switching and cognitive performance. Then, to link the states to periods of flexibility and stability in cognition, we further analyzed states that were particularly active during the maintenance or encoding phases of the tasks. We investigated how these states were influenced by task demands, such as cognitive load. Patterns of state dynamics observed in a third dataset, which included distractors, provided further support for these conclusions.

Finally, we explored the causal mechanisms relating synchronized states to behavioral responses. One such mechanism could be the control of information flow through phase-amplitude-coupling between the low-frequency synchronization and high-frequency, information-carrying spikes[21,22]. To test our hypothesis, we designed an in-silico brain model which could simulate information flow on a global scale.

The whole-brain model had both an oscillatory control layer and a spiking layer where information was assumed to be carried. The nodes in each layer had biologically realistic connectivity, estimated with diffusion weighted imaging. The spiking layer was modelled with spike rate[23] while the oscillatory layer was simulated using a Kuramoto model, which has successfully been used to reproduce synchronized brain networks[24]. Synchronized networks were initiated by an oscillatory input from a simplified basal-ganglia-thalamocortical loop – a control mechanism which has been suggested previously[25,26].

With the model, we evaluated if synchronization in the oscillatory layer could influence the flow of information in the spiking layer through phase-amplitude coupling. We investigated the influence of both the spatial distribution and the frequency of the synchronization.

## Results
### Data

We analyzed two independent MEG-datasets to identify low-frequency synchronized networks supporting vsWM. The first dataset consisted of MEG and behavioral data from 83 participants in the Human Connectome Project (HCP, https://www.humanconnectome.org/) who performed a 2-back task during MEG scanning (Fig. 1a). For each 2-back trial, an image was shown and the participants had to report if the current image matched the image from two trials earlier. The participants pressed one button for match and one button for non-match.

The second dataset consisted of repeated measures from four participants[27] who performed two different vsWM tasks: WM-Grid (Fig. 1b) and Odd One Out (Fig. 1c). In WM-Grid, the participants had to remember the order of five or six dots that were sequentially presented on a grid. After the whole sequence had been presented, a number between 1 and 16 was randomized into each grid slot. The participants answered verbally by reporting the number belonging to each grid position in the sequence.

In Odd One Out, a sequence of five stimuli was presented in each trial. Each stimulus consisted of three shapes, of which two were identical and the third differed. The task was to remember the position of the odd shape for each stimulus and report these positions in sequential order. The participants answered using three buttons (1 – 3), corresponding to the three possible positions.

The dataset had six recordings of Odd One Out and seven recordings of WM-Grid for each subject, collected over a period of eight weeks. While the HCP dataset was suitable for studying inter-individual similarities, the 4-subject dataset was used to determine the intra-individual reliability over tasks and time.

### Frequency analysis

Both datasets showed peaks in their global synchronization patterns in the theta and alpha bands, even though the individual peaks differed slightly (Fig. 2a). Figure 2b, c illustrates the temporal dynamics of the average global synchronization within the two frequency bands for

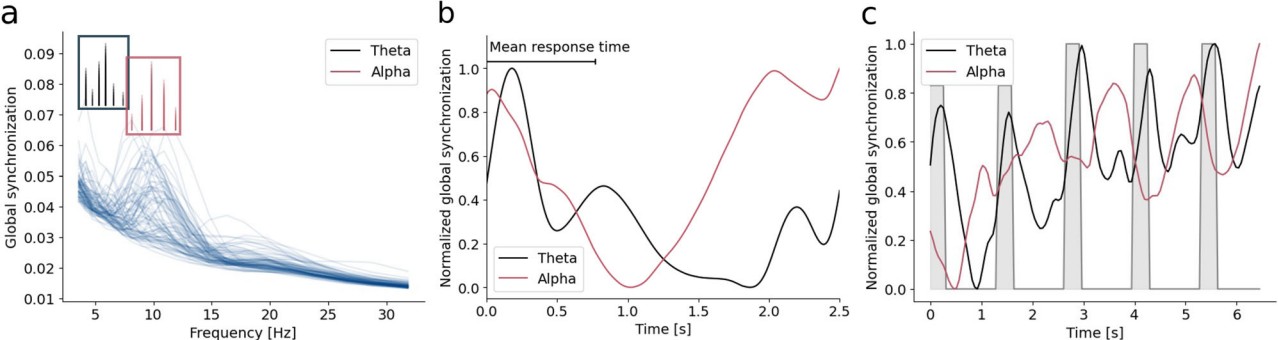

**Fig. 2 | Alpha and the theta oscillatory activity in the HCP and the 4-subject datasets. a** The average global synchronization as a function of frequency plotted for each subject in the HCP dataset. In the inserted diagrams, each bar marks individual peaks in synchronization, where a taller bar illustrates more subjects with a peak at that frequency. **b** The average synchronization across the 2-back trials. **c** The synchronization averaged across both the WM-Grid task and the Odd One Out task. The grey shaded regions mark the stimulus presentation periods. Source data are provided as a Source Data file.

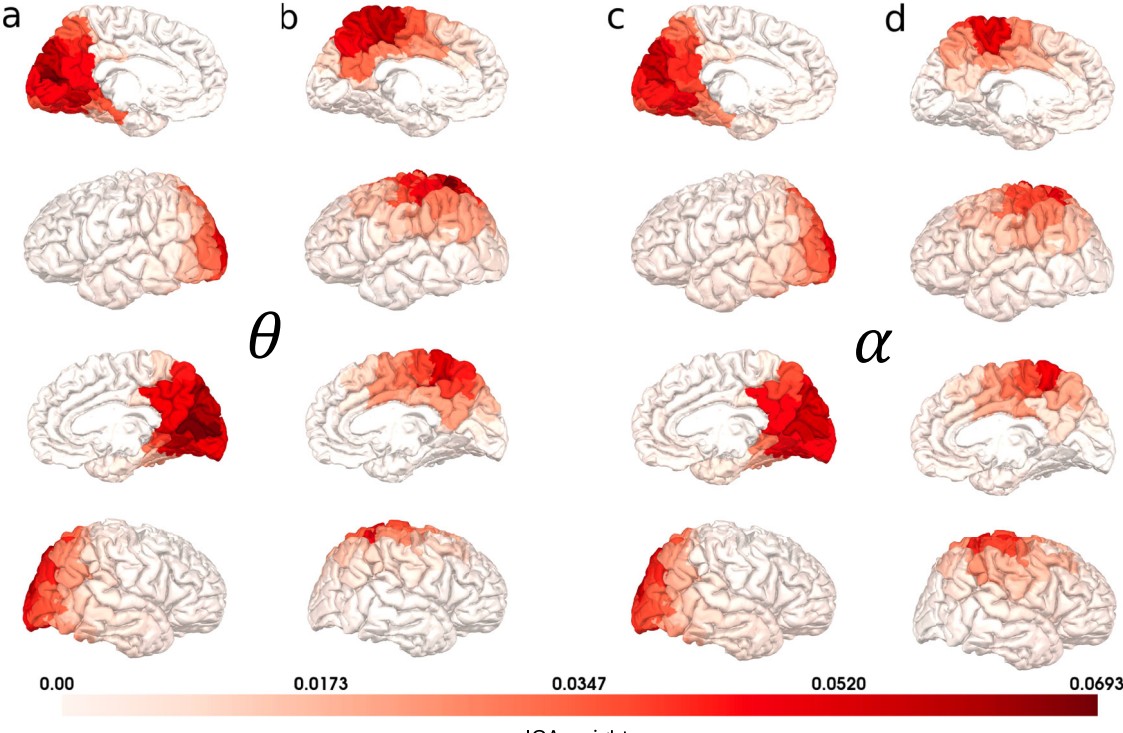

**Fig. 3 | The networks identified using ICA on wavelet data from the HCP dataset. a** The posterior theta network, (**b**) the dorsal theta network, (**c**) the posterior alpha network, and (**d**) the dorsal alpha network. Source data are provided as a Source Data file.

both tasks. Synchronization was calculated using the imaginary part of the phase-locking value[28].

### Identification of networks

We used ICA to separate independent signals in filtered trial data from the alpha and theta bands within each subject. For the 4-subject dataset, the trial data consisted of the presentation period (0–6.5 s) and for the HCP dataset, we used the whole trial (0–2.5 s). We identified two independent signals within each frequency.

The linear ICA operator was used to determine the brain regions generating the signal. The operator consisted of a weight vector with 200 weights, corresponding to 200 cortical regions. Regions with higher weights contributed more to the signal. These operators defined the spatial locations of the synchronized networks band (see Supplementary Fig. 1 for comparison between the global and the network specific raw signals).

In the theta band, we found one posterior and one dorsal network, primarily covering the occipital/posterior-parietal and parietal/posterior-frontal cortices, respectively (Fig. 3a, b). Within the alpha frequency band, we identified one posterior and one dorsal network with similar spatial distributions as for the theta networks (Fig. 3c, d).

In the 4-subject dataset, we also identified a fifth network of frontal theta synchronization, but it emerged over later sessions and was not present in the initial two scanning sessions. We therefore assumed that it arose due to repeated practice, and it was not investigated further in this study.

To assess the reliability of the networks, we used the 4-subject dataset. We compared the similarity of each network type over either sessions, tasks, or subjects, while controlling for the other two variables. For instance, when investigating network similarity across sessions, we only compared networks generated from the same subject and task. Not all networks were identifiable within each subset of the

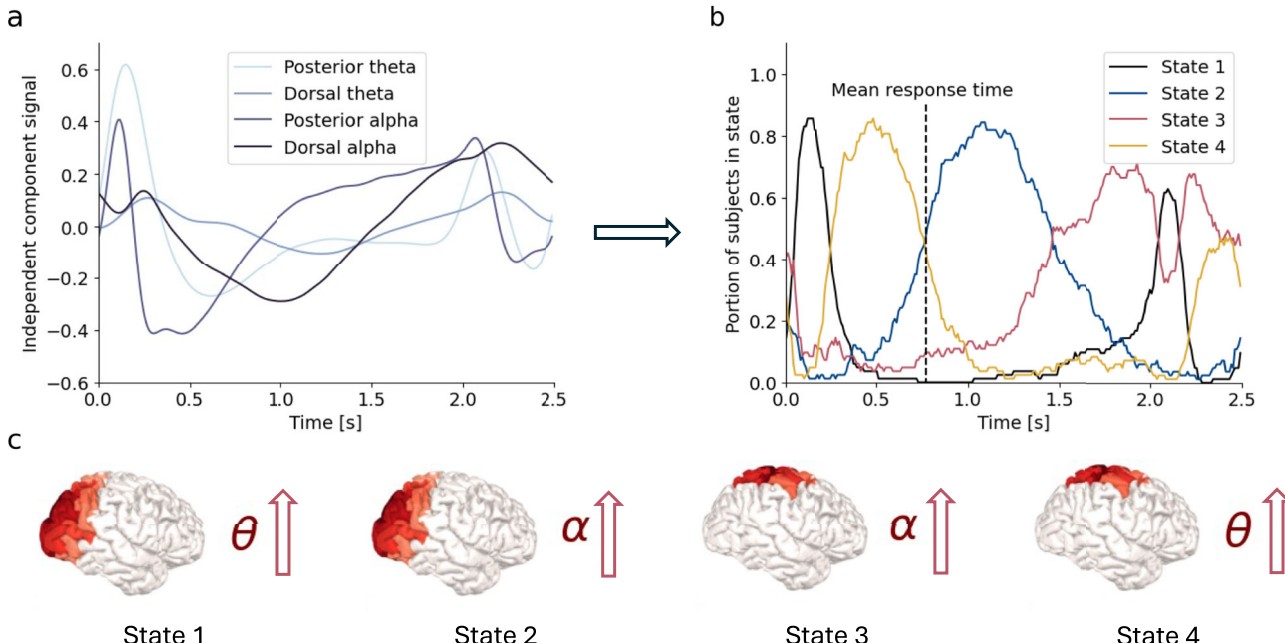

**Fig. 4 | The k-means clustering performed on the timeseries of the independent components in the HCP dataset.** For every subject, each time point is classified into one of four states: State 1 (posterior theta dominating), State 2 (posterior alpha dominating), State 3 (dorsal alpha dominating), and State 4 (dorsal theta dominating). **a** The signals of the independent components (i.e., network strengths). **b** The portion of subjects classified into each state at each timepoint. **c** The networks characterizing each state. Source data are provided as a Source Data file.

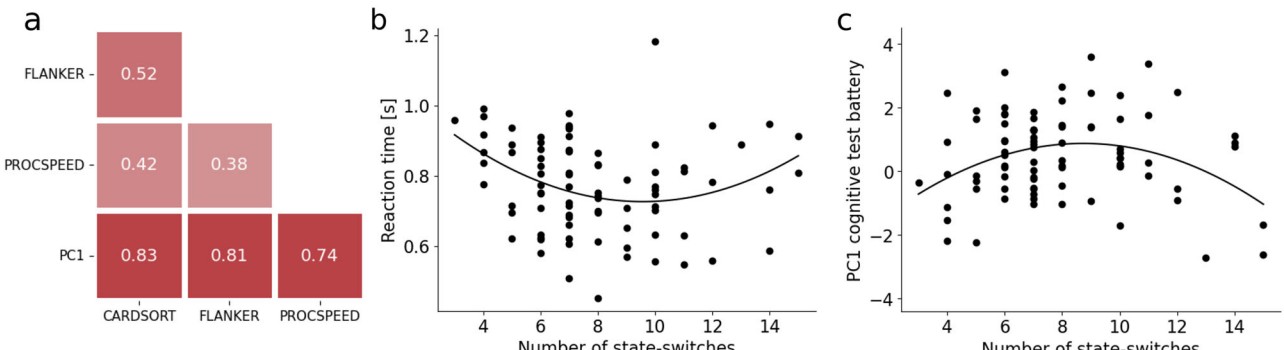

**Fig. 5 | The relationship between state-switches and performance in the HCP dataset. a** The correlation between the tasks in the test battery: Flanker's Inhibitory Control and Attention Task, Dimensional Change Card Sort, Pattern Comparison Processing Speed, and the first principal component (PC1) of a principal component analysis performed on the three tasks. **b** The reaction time on the 2-back task as a function of the number of state-switches. **c** PC1 as a function of the number of state-switches. Source data are provided as a Source Data file.

data, and in total we could identify 63% of all networks when the data had been partitioned into task and session for every subject.

For the comparison we used cosine similarity, which quantifies the similarity between two vectors on a scale from −1 to 1 (see Methods for more details). In this context, the vectors represented the ICA weight vectors that defined the networks. The analysis revealed a significant reliability across both task ($t = 37.1$, df $= 154$, $p < 10^{-78}$), session ($t = 13.2$, df $= 77$, $p < 10^{-21}$), and subjects ($t = 23.7$, df $= 58$, $p < 10^{-31}$) (Tab. 1). Here, a one-sided t-test was used to test if the mean cosine similarity was significantly larger than zero.

### Temporal clustering of networks into states
To investigate the interindividual similarities in the network strengths over time, we used k-means clustering and classified each time point into one of four states depending on the current network activities (Fig. 4a, b) (see Supplementary Fig. 2 for complete visualization of state generation pipeline). We found that the subjects alternated between four distinct

states, where each state could be characterized by a dominating network: state 1 – increased posterior theta; state 2 – increased posterior alpha; state 3 – increased dorsal alpha; and state 4 – increased dorsal theta (Fig. 4c). In the HCP, we found that at most, almost 90 percent of the subjects were in the same state at the same time, and on average, 65 percent of the subjects were in the dominating state at a given time point

### State-switches and cognitive performance
Next, we investigated how the states were related to cognitive performance. Here, we used two behavioral measures in the HCP dataset. First, we examined the performance during scanning in the 2-back task. Since there was a ceiling effect in accuracy where 83% of the subjects had an accuracy above 90%, we chose reaction time as our outcome measure. Second, we examined the performance in a separate battery of three cognitive tests: a card sorting task which tests mental flexibility, a flanker's task to test inhibitory control, and a test of processing speed (Fig. 5a). These three tasks were chosen because they

reflect basic executive processes and because there was a high correlation between them, leading us to believe that the underlying task mechanisms largely overlapped. The three tests were subjected to a principal component analysis and the first component (PC1) was used as a shared, general performance measure.

To study the impact of the state dynamics, we defined a state-switching rate, which was the number of transitions participants made between states during the trial. We found a significant quadratic relationship between number of state switches and reaction time ($\beta_{S.D.} = 0.24, 95\% \, \mathrm{CI}[0.08, -0.41], t = 2.88, \mathrm{df} = 80, p = 0.0051$) with optimal performance for subjects with around nine state changes (Fig. 5b). Similarly, we found a quadratic relationship ($\beta_{S.D.} - 0.36, 95\% \, \mathrm{CI}[-0.58, -0.13], t = -3.18, \mathrm{df} = 80, p = 0.0021$) between PC1 and the state-switching rate (Fig. 5c). Again, nine was the optimal number of state switches.

In all, these results suggested that cognitive performance is associated with the control of state transitioning.

## Identifying states for encoding and maintenance

To relate the states to encoding and maintenance in vsWM, the 4-subject dataset was used. In contrast to the 2-back task, the cognitive processes in the WM-grid and Odd One Out tasks are time-locked to trial events. More specifically, when stimuli are presented, the subjects should be in an encoding state, and during the delay periods, the subjects should be in a maintenance state.

We found that, during the encoding periods, state 1 dominated by posterior theta was most active, while state 3 dominated by dorsal alpha was most active during the maintenance periods (Fig. 6a). To confirm that the systematic switch between these states could be explained by vsWM maintenance and encoding, we recreated the state analysis in the same subjects but for a control task. The control task had the same timings as WM-Grid and Odd One Out. However, instead of a vsWM task, it was a verbal recognition task where the subjects reported 'yes' if the letter Q was presented and 'no' otherwise (Fig. 6c). Compared to the vsWM tasks, the verbal recognition task did not show a systematic switching between states 1 and 3 (Fig. 6b).

To further test the relationships of the states with encoding and maintenance, we investigated the load dependency of time spent in states 1 and 3 (Fig. 6d, e). For every session and task, we extracted the average time spent in each state for the four subjects. We then regressed this on WM-load. During stimulus presentation, we found a significant linear decrease of the time spent in state 1 for WM-Grid, $\beta_{S.D.} = -0.79 (95\% \, \mathrm{CI}[-1.01, -0.57], t = -7.33, \mathrm{df} = 33, p < 10^{-7})$, where $\beta_{S.D.}$ is the standardized regression coefficient. We did not see this for Odd One Out ($\beta_{S.D.} = -0.22, 95\% \, \mathrm{CI}[-0.59, 0.16], t = -1.17, \mathrm{df} = 28, p = 0.25$), even though there was a slight similar trend.

During the delay periods, we found a significant quadratic relationship where state 3 was most active during maintenance at load 3 for both WM-Grid ($\beta_{S.D.} = -1.00, 95\% \, \mathrm{CI}[-1.19, -0.80], t = -10.33, \mathrm{df} = 39, p < 10^{-11}$) and Odd One Out ($\beta_{S.D.} = -0.86, 95\% \, \mathrm{CI}[-1.14, -0.58], t = -6.19, \mathrm{df} = 33, p < 10^{-6}$). This coincided with an overall ceiling effect in alpha synchronization strength for load 3 and above, but without change in peak frequency (Supplementary Fig. 3).

## The control of posterior theta

The analysis of the WM-Grid task suggested that state 1 is down-regulated as the cognitive load increases, implying that it plays a role in input gating to vsWM. However, it is also possible that this decrease can be explained by the repeated exposure to a stimulus rather than an intrinsic regulatory mechanism. To disentangle this, we reanalyzed data from a third dataset consisting of a vsWM task with distractors[29]. Here, 13 subjects had to remember the rotation of either four bars presented in sequential order or, in the case of a distractor trial, the

rotation of only the first and forth bars presented (Fig. 7) (see ICA networks in Supplementary Fig. 4).

If state 1 were internally regulated, there would be differences in state duration during stimulus presentation between distractor and no distract trials, even though the stimuli were identical. Specifically, we expected that the time spent in state 1 would be shorter for stimuli 2 and 3 during the distractor trials. Moreover, according to the WM-Grid analysis, time spent in state 1 decreased with increasing cognitive load (Fig. 6d). Therefore, we predicted that the time spent in state 1 during stimulus 4 would be longer for distractor trials compared to no distractor trial as the cognitive load was two instead of four.

To test this, we calculated the time spent in state 1 during presentation periods for stimuli 2 (1.5–2.0 s), 3 (2.5–3.0 s), and 4 (3.5–4.0 s). We then calculated the difference between the two conditions within each subject (Fig. 7). That is, if the difference between distractor and no distractor trials was in the expected direction, this was recorded as a positive difference. If it was in the unexpected direction, it was recorded as a negative value. For each of the 13 subject we therefore got an overall trend for the three stimuli together. With a one-sided paired t-test, we found that the difference was significantly larger than zero (mean difference for all three time periods $= 128.2 ms$, $95\% \, \mathrm{CI}[43.8, \inf], t = 2.73, \mathrm{df} = 12, p = 0.009$, Cohen's delta (d) $= 0.79$).

## In-silico simulations of low-frequency networks for information routing

The analysis of MEG data suggested that large oscillatory networks form functional brain states. Yet, we so far lack a mechanistic explanation for how these states influence neural activity. We hypothesized that low-frequency networks can control where information in high-frequency spiking is routed through phase-amplitude coupling. To test this hypothesis, we used an in-silico brain model.

Apart from gaining a better mechanistic understanding through modelling, the simulations allowed us to measure information flow in terms of transfer entropy. Transfer entropy has many advantages, such as the ability to quantify non-linear information flow. However, it requires relatively large amount of continuous data, which is not well adapted to the quick state changes in real data. Furthermore, in the simulated data, we do not have any issues with low signal-to-noise ratio which is otherwise a problem when analyzing high-frequency spiking in MEG data[30].

The in-silico model consisted of 200 cortical nodes corresponding to regions in the Schaefer atlas[31] and two basal-ganglia-thalamus nodes, one in each hemisphere (Fig. 8a). The node connectivity as well as distances between nodes were constrained by structural data that were acquired from ten individuals in the MICA-MICs MRI dataset[32]. All simulations were first run within each single subject, and the results were then averaged across the ten subjects.

Each node in the model had two layers: a spiking layer and an oscillating layer. The spiking layer was modelled with spike rate[23], and the oscillatory layer was created using a Kuramoto model[24] consisting of coupled oscillators. The communication between nodes within each layer depended on signal strength as well as the distances between nodes. Furthermore, the spiking activity was influenced by the oscillatory activity (but not the other way around) through phase-amplitude coupling. More specifically, spiking activity was enhanced during the troughs and suppressed during the peaks of the oscillations, as has previously been shown experimentally[15,33].

In the simulations we evaluated 1) if synchronization could emerge within the areas identified in the experimental data, 2) if the synchronization could be used to facilitate information flow, and 3) if the information flow depended on synchronization frequency.

## Generating networks using Thalamic input

We hypothesized that coordinated input from thalamus could work as a mechanism for synchronizing the cortical nodes. Therefore, we

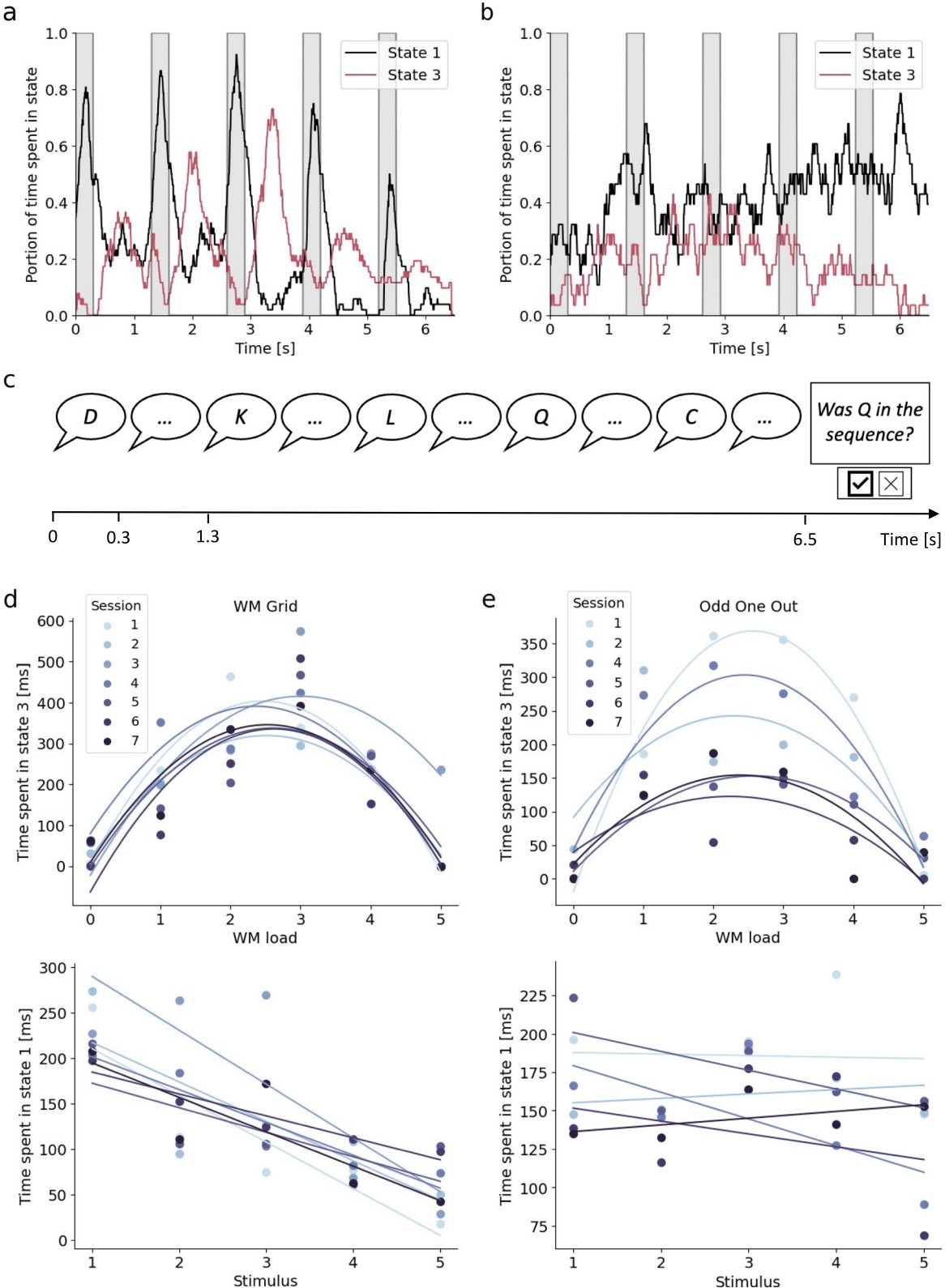

**Fig. 6 | Investigation of the relationship between WM load and the time spent in states 1 and 3 in the 4-subject dataset. a** The portion of time spent in states 1 and 3, averaged across sessions, subjects, and tasks. The shaded areas mark stimulus presentation periods, and the non-shaded areas mark the delay periods. **b** The portion of time spent in states 1 and 3 during the verbal recognition task, averaged across subjects and sessions. **c** The verbal recognition task used as a control. Letters are presented verbally in sequential order, and at the end of the presentation, participants responded 'yes' if the letter Q had been presented and 'no' otherwise. **d** The time spent in states 1 and 3 during the WM-Grid task. Each dot represents the mean time for each session. Above: the time spent in state 3 during the delay periods. Load 0 is the 1000 ms before presentation of the first item, and load 5 is the 1000 ms delay period after the presentation of the fifth item. Below: the time spent in state 1 during the 300 ms presentations of each stimulus 1–5. **e** The time spent in states 1 and 3 during the Odd One Out task. All analysis and time periods are identical to those of WM-Grid (apart from the Odd One Out task not being included for measurement session 3). Source data are provided as a Source Data file.

specifically increased the strength of the oscillatory thalamic activity (Fig. 8b) to either the posterior regions (Fig. 3a) or the dorsal regions (Fig. 3d). Next, the synchronization between cortical nodes was calculated using phase-locking value[34].

The thalamic input was sufficient to selectively increase synchronization in dorsal or posterior regions. Averaged over subjects, the difference between synchronization inside compared to outside the network was 63% (standard deviation (SD) = 15.7%) in the posterior network and 62% (SD = 9.3%) in the dorsal network when they received thalamic inputs. Using a one-sided t-test we found that this was significant within both networks (posterior network: mean synchronization difference = 0.08, 95%CI[0.06, inf], $t = 10.20$, df = 9, $p < 10^{-5}$, $d = 3.40$; dorsal network: mean synchronization increase = 0.09, 95% CI[0.06, inf], $t = 14.81$, df = 9, $p < 10^{-7}$, $d = 4.94$) (Fig. 8c-d).

### The influence of synchronization on information flow
Next, we evaluated how the synchronization influenced the flow of information in the spiking layer. We specifically hypothesized that

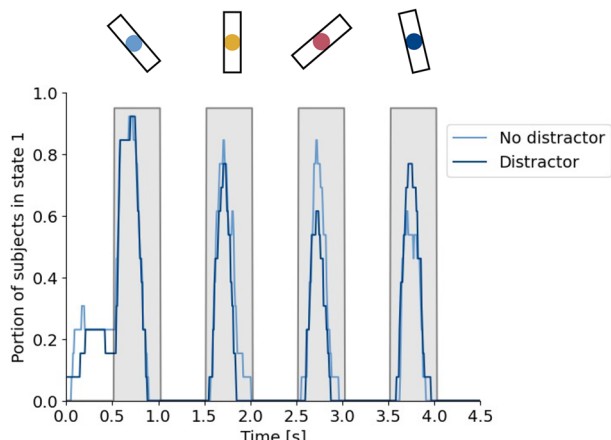

**Fig. 7 | The time spent in state 1 in distractor trials compared to no distractor trials.** Before the trial starts, a cue was shown to indicate if the trial was a distractor or a no distractor trial. At 0.5, 1.5, 2.5, and 3.5 s, a stimulus was shown for 0.5 s, followed by a 0.5 s delay period. The shaded areas mark the presentation of the stimuli. The subject was to remember the rotation of the bar and the color of the dot in the middle of the bar. After presentation of the four items, one of the four colors were shown, telling the subject to recall the rotation of the matching bar. Source data are provided as a Source Data file.

synchronization would facilitate information transfer from nodes inside the synchronized network. Thus, we increased the external input ($I$ in Eq. 2, see Methods) to a spiking unit within the network and quantified the resulting information flow to other nodes using transfer entropy. Again, all simulations were run within single subjects and subsequently averaged across the ten.

We examined four conditions: stimulation to either primary visual cortex (V1) or the intraparietal sulcus (IPS) under either dorsal or posterior synchronization. We chose these specific nodes due to their involvement in early visual processing and vsWM maintenance[35] respectively. The synchronization frequency was set to 10 Hz in all four conditions.

As we hypothesized, we found that the total transfer entropy from V1 to the rest of the brain was 201 % (SD = 80%) larger during posterior synchronization compared to dorsal synchronization, whereas total transfer entropy from IPS to the rest of the brain was 207 % (SD = 28%) larger during dorsal synchronization (Fig. 9a) (one sided paired t-test, posterior network: mean difference in transfer entropy = 0.018, 95%CI[0.013, inf], $t = 7.83$, df = 9, $p < 10^{-4}$, $d = 2.61$; dorsal network: mean difference in transfer entropy = 0.021, 95% CI[0.019, inf], $t = 26.88$, df = 9, $p < 10^{-9}$, $d = 8.88$).

Furthermore, the increase occurred most strongly within the synchronized networks themselves (two sided paired t-test; synchronized posterior network vs rest of the brain: mean difference = 0.018, 95% CI[0.013, 0.023], $t = 8.60$, df = 9, $p < 10^{-4}$, $d = 2.87$; synchronized dorsal network vs rest of the brain: mean difference = 0.019, 95% CI[0.016, 0.021], $t = 15.95$, df = 9, $p < 10^{-7}$, $d = 5.32$) (Fig. 9b).

### The influence of frequency on information flow
After showing that synchronization facilitated information flow in the model, we further investigated how frequency could modulate this flow. In the experimental data, each identified region had networks both in the theta and alpha range and therefore, we again tested four simulation conditions: stimulation to V1 under either posterior theta (6 Hz) or posterior alpha (10 Hz) synchronization, and stimulation to the IPS under either dorsal theta or dorsal alpha synchronization.

The frequency of posterior synchronization did not affect transfer entropy (two-sided paired t-test, $p = 0.85$), but in the dorsal regions, the transfer was overall higher under theta compared to alpha synchronization (two-sided paired t-test, mean difference = 0.011, 95% CI[0.008, 0.013], $t = 11.64$, df = 9, $p < 10^{-6}$, $d = 3.88$) (Fig. 9c). However, even though there was an overall decrease in information transfer during dorsal alpha synchronization, especially to the contralateral hemisphere, information transfer was also increased to some

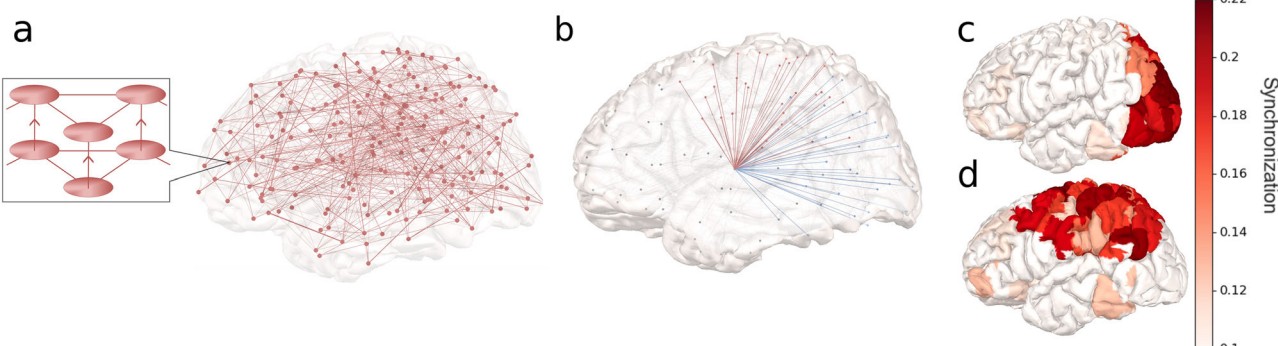

**Fig. 8 | Model architecture and generation of networks in the model. a** The 202-node network of the model. Each node represented either a cortical or sub-cortical region. Each node had two units, one unit with spiking activity and one unit with oscillatory activity. The oscillatory layer influenced the spiking layer within each node through phase-amplitude coupling. The connectivity within each layer was estimated with diffusion weighted imaging. **b** Synchronization was created by increasing the activity from the thalamus to regions in the cortical network to be synchronized, which was either the posterior theta (blue) or dorsal alpha (red) network. **c** Model synchronization in the posterior network after increased thalamic input to the network. Synchronization was calculated using phase-locking-values. **d** Model synchronization in the dorsal network.

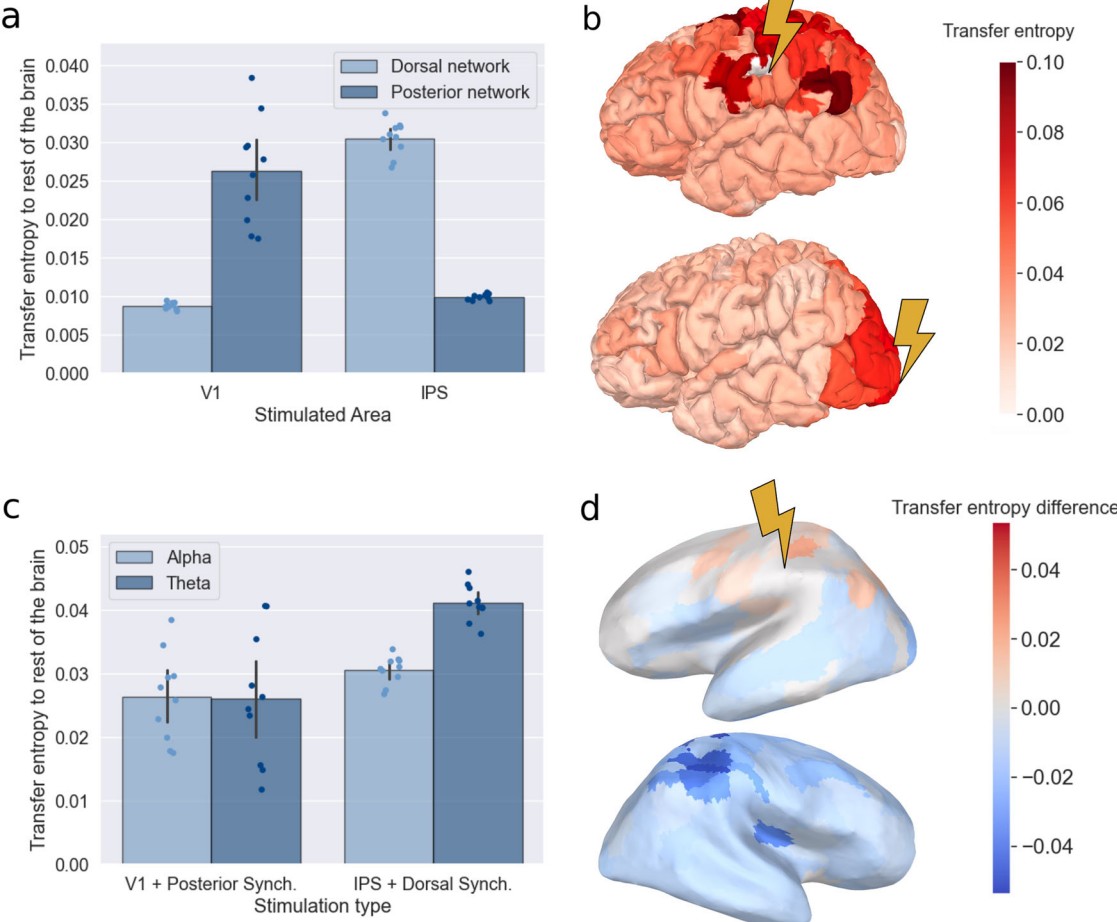

**Fig. 9 | Information transmission in the model calculated as transfer entropy across the ten subjects. a** The transfer entropy from stimulated area (V1 or IPS) to the rest of the brain. The dots show the transfer entropy of individual subjects, and the error bar shows the 95% confidence interval of the mean transfer entropy. The mean transfer entropy is the center of the error bars. **b** On top, the transfer entropy during IPS stimulation and synchronization in the dorsal network. Below, the transfer entropy during V1 stimulation and synchronization in the posterior network. **c** The transfer entropy from V1 under posterior synchronization and IPS under dorsal synchronization, for both theta and alpha oscillations. The dots show the transfer entropy of individual subjects, and the error bar shows the 95% confidence interval of the mean transfer entropy. **d** The mean difference in transfer entropy from IPS to other brain regions under dorsal alpha vs dorsal theta synchronization. Insignificant changes are marked as white.

areas. These areas included the frontal eye field and posterior cortex of the stimulated hemisphere (Fig. 9d). Significant differences in transfer entropy were calculated within each node separately and multiple comparison corrected using False Discovery Rate (FDR).

## Discussion

We found that, during vsWM tasks, brain states can be characterized by large-scale, synchronized networks in the alpha and theta bands. Optimal control of the switching between these states was associated with cognitive performance, both during MEG scanning and in separate cognitive tests. We further identified a state dominated by dorsal alpha synchronization as a maintenance state and another state dominated by posterior theta synchronization as an encoding state. Both these networks were modulated by cognitive load. Simulations further demonstrated that synchronized networks could facilitate information transfer between regions, promoting information to be differentially routed based on frequency and spatial location (see Supplementary Fig. 5 for schematic overview of the results).

This study focused on visuospatial memory and the core types of information processing required for such tasks, namely visuospatial stability and flexibility. We believe that the identified brain states represent these fundamental processes.

For example, similar activation patterns to the dorsal alpha network are often associated with vsWM maintenance[6,9,36,37], and stronger alpha band synchronization within these areas has been linked to higher WM capacity, both inter-individually[38–41] and intra-individually[27]. However, dorsal alpha synchronization can also be associated with other domains such as top-down visuospatial attention[42,43], pre-saccade onset in natural reading[44] and motor planning[45], all of which rely on stable spatial representations.

Our experimental data further strengthen the link between dorsal alpha and visuospatial stability. The WM-Grid task, which heavily relies on spatial representations, elicited a stronger dorsal alpha signal than the Odd One Out task, where the sequence can also be remembered as numbers (1–3) marked on the response buttons. Additionally, the highest dorsal alpha signal occurred at a cognitive load of three, consistent with the number of items humans can hold in memory without making noticeable errors[46]. Beyond this load, additional processes such as compression – relying on patterns in the data[47] – may supplement visuospatial stability for maintaining items in WM.

In terms of flexibility, occipital event-related responses in the theta band – early after stimulus onset and before the P3 component – have been associated with visual processing[48–51]. These may in turn be modulated by ongoing intrinsic oscillations, which have been shown to

influence visual detection threshold[52] and reaction times to visual stimuli[53].

In the experimental data, we also found evidence for the internal regulation of the posterior theta state. For example, in the distractor task, the temporal dynamics of the posterior theta state were modulated differently for distractor and no distractor trials, even though the stimuli were identical. Moreover, while the state was downregulated by load in the WM-Grid task, this downregulation was not significant in the Odd One Out which required an additional step of visual processing compared to WM-Grid.

For encoding and maintenance, we identified two distinct states, but in total we found four states that differed in spatial location and/or frequency. For the two dorsal states, it was possible to investigate differences between the dorsal alpha state and its theta counterpart using the computational model. The model revealed that the alpha state prioritized frontoparietal information transmission within the hemisphere, whereas the theta state transmitted more information across hemispheres. Interhemispheric functional connectivity has been linked to vsWM manipulation[54] suggesting that dorsal theta may play a greater role in manipulation rather than maintenance. This hypothesis aligns with the temporal dynamics of the theta state, which occurred a few hundred milliseconds after the posterior theta state (see Supplementary Fig. 6).

The two posterior states are more challenging to interpret. We identified posterior theta to be dominating during encoding; however, the computational model found no significant differences in information processing between the theta and the alpha networks. To add to this, occipital oscillations linked to visual processing have been observed both in the theta and the alpha band[48,51,55,56]. Yet, there are also clear differences. For example, only alpha oscillations have been associated with distractor suppression[57,58] and eyes-closed resting state. Further research is needed to clarify the distinctions between the posterior frequencies and continued development of computational models could provide deeper insights into these dynamics.

Another often described neural correlate of WM is sustained frontal theta oscillations[9]. However, we did not find evidence of this in our experimental data. In general, vsWM maintenance is not commonly associated with spectral peaks in frontal theta. Instead, frontal theta oscillations recorded with M/EEG in humans appear more often during maintenance of verbal information (see Table 1 in Roux and Uhlhaas 2014[9]), and while the networks we identified may not be specific to WM, they may be specific to visuospatial processing.

In monkeys, intracranial recordings have provided evidence for a frontal theta component also during vsWM[59] and it is possible that the poor signal-to-noise ratio from frontal sensors in MEG limits the frontal theta signal in human studies. Alternatively, the extensive training required in primate experiments might lead to the emergence of different signals. For instance, we identified a mid-frontal theta component that became apparent during later sessions in a 4-subject dataset. This component may be less related to the fundamental states of visual perception and cognition described here.

Finally, the relationship between state switching and cognitive performance has a U-shaped curve. Interestingly, this resembles the characteristic U-shaped relationship between dopamine levels and cognitive performance, where too little dopamine leads to reduced cognitive stability and too much dopamine leads to reduced cognitive flexibility[60,61]. The mechanisms underlying the control of state transitions remain unclear, but previous research suggests that fronto-basal ganglia interactions could be involved. It has for example been suggested that these activations help to stabilize WM representations[26] and that they are important for flexibility during cognitive tasks[62]. In monkeys, intrinsic theta oscillations generated by the pulvinar have also been implicated in the regulation of spontaneous changes in visual flexibility and stability[63].

To conclude, our study highlights the importance of fundamental oscillatory states of flexibility and stability in vsWM, where the control of these states was related to cognitive performance. Based on the results of the computational simulations, we further show that the states have the potential to influence information flow through spatial location and oscillatory frequency.

## Methods

### HCP dataset

**Participants and scanning sessions.** The HCP contains functional and structural MRI, diffusion weighted imaging, and genetic and behavioral data from more than 1200 young adults. A subset of 83 subjects (45 men) performed a WM task in a MEG scanner. Of these, 17 subjects were in age group 22–25 years, 33 were in age group 26–30, and 33 were in age group 31–35 (subject specific age and gender provided in the source data). The subjects were scanned during a three-hour session that also included resting state, a language processing task, and a motor task. Each task was recorded in two runs.

**Task.** The WM task was an N-back task (Fig. 1a) with two different load conditions. The first condition was match-to-sample (0-back), where subjects had to report whether the presented image corresponded to a predefined target image. In the second condition (2-back), subjects reported if the current image matched the image from the stimulus presented two steps earlier. The images were either tools or faces. Stimuli presentation lasted for 2000 ms followed by a 500 ms fixation period. Each block was composed of ten stimuli that could either be all tools or all faces. There were eight blocks per condition and stimulus combination.

**MEG acquisition and preprocessing.** The MEG data, recorded from a whole head MAGNES 3600 (4D Neuroimaging) system, had already been preprocessed. Bad channels had been identified by computing the correlation and ratio variance of each channel with respect to their neighbors, and bad segments had been detected using z-score metrics and an iterative approach based on ICA. This data was excluded from the final dataset. Then, the data had been decomposed into independent components using ICA, and the components classified as non-brain components had been discarded. The data had also been down sampled to 508.63 Hz. Finally, the data had been epoched into single trials ranging from −150 ms to 2650 ms. Here, time windows of 150 ms had been added to the begin and end of the trial to account for edge effects from later filtering.

**MRI acquisition.** T1-weighted structural MRIs ($1 \times 1\,m^2$) were acquired with a Siemens 3 Tesla (Connectome Skyra).

**Resources for HCP analysis.** All subsequent data analysis performed on the HCP dataset was enabled by resources in projects NAISS 2023/22-814 and NAISS 2023/23-524 provided by the National Academic Infrastructure for Supercomputing in Sweden (NAISS) at UPPMAX, funded by the Swedish Research Council through grant agreement no. 2022-06725.

**Table 1 | The median cosine similarities between networks across sessions, tasks, and subjects**

| Networks | Similarity across sessions | Similarity across tasks | Similarity across subjects |
|---|---|---|---|
| Posterior theta | 0.60 (0.23) | 0.69 (0.17) | 0.48 (0.21) |
| Dorsal theta | 0.53 (0.20) | 0.75 (0.23) | 0.41 (0.21) |
| Posterior alpha | 0.67 (0.21) | 0.87 (0.21) | 0.60 (0.19) |
| Dorsal alpha | 0.75 (0.21) | 0.80 (0.25) | 0.36 (0.25) |

Standard deviations are shown in parenthesis.

## 4-subject dataset

**Participants and scanning sessions.** The dataset included two men and two women aged 21, 21, 22 and 26. All participants provided written consent to the study and the study design was approved by the Swedish Ethics Committee. Each participant was scanned on seven different occasions over a period of eight weeks, where the scanning occurred on days 1, 2, 4, 10, 19, 29 and 39. During the remain days, they participated in a cognitive training program from home[29].

**Tasks.** In the scanner, the participants performed two WM tasks and one control task. The first task was WM-Grid (Fig. 1b). Here, a 4 × 4 grid was displayed on screen. Positions on the grid lit up, and the task was to remember the position of the stimuli in sequential order. The task contained 40 trials with five stimuli and 40 trials with six stimuli. Stimuli were presented for 300 ms with 1000 ms delay periods.

The second task was Odd One Out (Fig. 1c). Here, each stimulus in the sequence contained three shapes, of which two shapes were identical while the third differed. The task was to remember the positions of the odd shape (left, middle, or right). Odd One Out included 40 trials with five stimuli. Again, stimulus presentation lasted 300 ms with 1000 ms delays.

The third task was a verbal recognition task used as a control (Fig. 5c). Here, a sequence of 5 or 6 letters were presented, with 1300 ms between the start of each letter presentation. This gave us a similar temporal dynamic for the control task with respect to the two vsWM tasks. The task was to listen for the letter Q and, at the end of the sequence presentation, press 'yes' if Q was included in the sequence and 'no' otherwise.

**MEG acquisition and preprocessing.** MEG data was acquired with a 306-channel whole-head MEG system (Elekta Neuromag TRIUX) at the Swedish national facility for MEG, NatMEG (https://natmeg.se/). The sampling frequency was 1000 Hz. To reduce external artifacts, we applied Temporal extension of signal space separation (tSSS) with the software MaxFilter, and to reduce artifacts from heartbeats and eye movements, we applied ICA. Finally, we used notch filtering to remove line noise at 50 Hz and its harmonics. ICA and notch filtering was performed using the python package MNE-Python (https://doi.org/10.5281/zenodo.592483). Finally, we separated the data into epochs containing a trial and a 300 ms time window on each side to account for edge effects.

**MRI acquisition.** We acquired T1-weighted structural MRIs (1 x 1 m²) with a 3 Tesla scanner (GE SIGNA Premier).

## Distractor dataset

**Participants and scanning sessions.** The distractor dataset included structural MRI and MEG task data from 17 participants aged 21–41[29]. Four subjects had to be removed due to the inability to identify the theta networks. They were removed before any additional state analysis was performed. This resulted in 13 subjects for the final analysis.

**Task.** The WM task was a sequential vsWM task with distractors. Each trial started with a pre-cue, indicating if the trial had distractors or not. 50 percent of the trials were distractor trials. Thereafter, four bars were shown in sequential order for 500 ms, separated by 500 ms delays. Each bar had a random orientation and was marked with a colored dot. Both the orientation and color were to be remembered. On the distractor trials, bars 2 and 3 were distractors and did not have to be remembered. The presentation was followed by a 750 ms fixation period. Then, the fixation dot changed color indicating which of the four bars should be recalled. After another 750 ms, the subjects reported the orientation of the bar which was to be recalled. There were 400 trials in total in blocks of 40.

**MEG acquisition and preprocessing.** The MEG data had been recorded with the same equipment as for the 4-subject dataset and was already preprocessed. Temporal extension of signal space separation (tSSS) had been applied with MaxFilter. ICA had been used to remove eye movements and muscles artefacts, and a semi-automated method had been used to remove heartbeat. Both were done with the MATLAB package Fieldtrip[64]. The data had also been epoched into trials and down sampled to 250 Hz. The trials started at −2000 ms (with the pre-cue) and ended with the recall at 6000 ms, where the first bar was presented at 500 ms (see Fig. 6)

**MRI acquisition.** T1-weighted structural MRIs (1 x 1 m²) were acquired with a Siemens 3 Tesla.

## Common processing of MEG data for all datasets

**Anatomical reconstructions and parcellation.** We used FreeSurfer (http://freesurfer.net/) for anatomical reconstructions. The reconstructions were then parceled into 200 regions defined by the seven network Schaefer atlas[31].

**Source reconstruction.** We projected the MEG-signals from sensor space to source space using the dSPM method for minimum norm estimation (MNE) in MNE-Python[65]. Here, we used evenly distributed dipole sources with 5 mm intervals. They had fixed orientations normal to the pial surface. For the 4-subject dataset, the noise covariance matrix was estimated using the first 1000 ms of each trial before the presentation of the first stimulus. For the HCP and distractor datasets, all task data was used to construct the noise covariance matrices. The noise covariance matrices were also regularized. To collapse the source time series into parcel time series, we first optimized the collapse-operator to maximize the similarity between simulated data and its reconstructed signal[66].

**Morlet filtering.** We performed wavelets transformation using Morlet wavelets at 6 and 10 Hz with MNE-python[67]. The number of cycles per wavelet was set to 5 and the filtered data was down sampled to five times the center frequency. Finally, we removed the time windows of either side of the trial.

## Networks

**Creating networks.** We extracted the absolute value of the filtered data from its complex form, and normalized it within frequency, session, subject and task. To separate independent sources in each frequency, we applied ICA using the python toolbox scikit-learn[68]. The appropriate number of components used in the algorithm differed between participant and frequency but were always between two and five components. The ICA-timeseries were averaged across trials within session, subject, and task. Finally, we concatenated the ICA-timeseries such that each timepoint had one value for each independent component. We used k-means clustering to classify each timepoint into one of four clusters, again with scikit-learn. The clustering was done separately within each dataset. Examples of data before filtering and ICA is shown in S4 for comparison.

Here, it should be noted that the 4-subject dataset contained an additional frontal theta component, which emerged over time. This component was included in the clustering analysis meaning that an additional fifth cluster was found, dominated by high frontal theta. However, this component was not included in the results, as it seemed to emerge due to repeated exposure to the tasks and was therefore not relevant for this analysis.

**Comparing networks.** To compare the networks, we used cosine similarity. The cosine similarity between vectors $\mathbf{u}$ and $\mathbf{v}$ is defined as the dot product between their norms: $\bar{\mathbf{u}} \cdot \bar{\mathbf{v}}$. It will have a value of 0 if the vectors are orthogonal and independent, and a value of 1 if the

vectors point in the same direction. In this case, the vectors were the ICA weight vectors, inferred for each subject, session, and task.

For comparing similarity across sessions, we kept subject and task constant, hence only comparing ICA weights from the same subject and task. For similarity across tasks, we kept session and subject constant, and likewise, for similarity across subjects, we kept session and task constant.

## Computational modelling

**Structural data.** Structural data was acquired from the MICA-MICs dataset[32]. The dataset included predefined connectivity matrices between the 200 regions in the Schaefer atlas as well as seven subcortical regions in each hemisphere (nucleus accumbens, amygdala, caudate nucleus, pallidum, putamen, thalamus, and hippocampus). Furthermore, the dataset also contained distance matrices, extracted from T1-weighted images, with estimated distances between all cortical and subcortical regions. The connectivity strengths were defined by the weighted streamline counts, estimated using multi-shell diffusion weighted images.

For this model, we kept all cortical nodes, removed the amygdala and the hippocampus, and summed the remaining subcortical connections up into one basal-ganglia-thalamus node. We did this because we assumed that the synchronization observed in the data was mainly a cortical phenomenon, working in interaction with the basal-ganglia-thalamus, but not a phenomenon that occurs between the basal-ganglia-thalamus nodes themselves. Hence, interactions between subcortical were treated as a black box.

It is well established that input from the cortex mainly goes to the basal ganglia, while the output to the cortex is thalamic. Therefore, we assumed that all incoming connections to our subcortical node went to the basal ganglia and all outgoing connections were thalamic.

**Kuramoto layer.** The evolving phases of the oscillators in the Kuramoto model were generated by the equation:

$$\frac{d\theta_n}{dt} = \omega + k\sum_{p=1}^{N} C_{np} \sin\left(\theta_p\left(t - \tau_{np}\right) - \theta_n(t)\right), n = 1, \ldots, N \quad (1)$$

where $\theta_n$ was the phase of node $n$, $\omega = 0.04 \cdot 2\pi$ was the natural frequency of the nodes, $k = 0.22$ was a global connectivity strength for the oscillating layer, $C_{np}$ was the connectivity strength between node $n$ and node $p$, and $\tau_{np} = 2 \cdot D_{np}$ was the time delay. $C$ was the connectivity matrix, normalized such that the incoming weights to a node added up to one, and $D$ was the distance matrix. The parameter values were based on previous work on the Kuramoto model in networks that resemble brain connectivity[24]. The delay parameter implicates that the average time it took for a signal to travel from one node to another was 23 ms. In the model, we also added white noise with variance 0.0025. For simulating the time series, we used Euler integration with a timestep of $dt = 0.1$ ms.

To generate the synchronized networks, the outputs from subcortical nodes were increased to the cortical regions within the network. In practice, this was done by increasing the values of the connectivity matrix $C_{np}$ for the Kuramoto layer but not the spiking layer. To modulate the synchronization frequency, we changed the oscillatory frequency of the subcortical nodes from the natural frequency $\omega$ to the desired synchronization frequency.

**Spiking layer.** Nodes in the spiking layer have the following dynamics:

$$\frac{du_n}{dt} = w\sum_{p=1}^{N} C_{np} u_p\left(t - \tau_{np}\right) - a \cdot u_n(t) + I_n(t) \quad (2)$$

where $u_n$ was the spike rate for node $n$, $w = 0.8$ was a global connectivity strength, $C_{np}$ was the connectivity strength between node $n$ and node $p$, $\tau_{np} = 2 \cdot D_{np}$ was the time it took for a signal to travel from node $p$ to node $n$, $a = 0.25$ was the intrinsic inhibition strength, and $I$ was an external current which could be used to initiate signals. Again $C$ and $D$ were the connectivity and distance matrices, and a timestep of $dt = 0.1$ ms was used for simulations. To perturbate the system at node $n$, $I_n$ was set to 1.

**Phase amplitude coupling.** In each node, the oscillatory layer influenced the spiking layer through phase amplitude coupling:

$$\frac{du_{n,PAC}}{dt} = u_n \cdot \left(-0.5\sin(\theta_n) + 0.5\right) \quad (3)$$

where $u_n$ was the spike rate and $\theta_n$ was the phase of the oscillation. This means that the spiking activity was multiplied with a value between 0 and 1.

**Calculating transfer entropy.** Transfer entropy from node $a$ to node $b$ with a time delay of $\Delta$ was defined by the following equation:

$$d\mathrm{TE}_{a\rightarrow b}(\Delta) = \iiint p_{a,b,b(\Delta)} \log\left(\frac{p_{a,b,b(\Delta)}}{p_{a,b}p_{b,b(\Delta)}}\right) dx_a dx_b dx_{b(\Delta)} \quad (4)$$

Here, $p_{a,b,b(\Delta)}$ was the joint probability distribution, defined numerically. To this end, we first normalized the time series separately for each node. Thereafter, we binned the timeseries into values between 0 and 20 and used those values to construct the joint distributions. Finally, we summed up $d\mathrm{TE}_{a\rightarrow b}(\Delta)$ for all values of $\Delta$ between timesteps 1 and 400, corresponding to 0.1–40 ms.

## Statistical measures

**Regressions.** All regressions were performed using the python package statmodels and the class OLS, which uses ordinary least square.

**Significance test.** All significance tests were performed using the python package scipy. For paired data we used the function ttest_1-samp and tested if the difference between the two conditions differed from zero. For independent samples, we used the function ttest_ind, testing for different means between the two samples. We used both one-sided and two-sided $t$-tests depending on whether or not we had a prior hypothesis. This is reported together with the results of each test.

**Effect sizes.** Standard coefficients were used to describe the effects of regressors. These were calculated by first normalizing the data (independent and dependent variables) before the regression.

When comparing two data samples, we used Cohen's delta. For paired data this was calculated as:

$$d = \frac{\mu_{\Delta X}}{\sigma_{\Delta X}}$$

$\mu_{\Delta X}$ is the mean difference between the two samples and $\sigma_{\Delta X}$ is the standard deviation of the difference. For independent samples we used:

$$d = \frac{\mu_x - \mu_y}{\sigma}$$

Where $\sigma$ is the pooled standard deviation:

$$\sigma = \sqrt{\frac{(n_x - 1)\sigma_x^2 + (n_y - 1)\sigma_y^2}{n_1 + n_2 - 2}}$$

## Reporting summary

Further information on research design is available in the Nature Portfolio Reporting Summary linked to this article.

## Data availability

Data from the Human Connectome Project is available in the S1200 release at the Human Connectome project website, https://www.humanconnectome.org/. Data from the 4-subject dataset is available at Open Science Framework (OSF), https://osf.io/8mvwy/. Data from the distractor dataset is available at Open Science Framework (OSF), https://osf.io/gu25f/?view_only=. Additional raw data for the structural matrices used for simulations can be found at the Canadian Open Neuroscience Platform (CONP) Portal, https://portal.conp.ca/dataset?id=projects/mica-mics. Source data for all figures and the table is provided in this paper. Source data are provided with this paper.

## Code availability

Code for the simulations can be found on GitHub: https://doi.org/10.5281/zenodo.15374274[69].

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

## Acknowledgements

Data were in part collected from NatMEG, The National infrastructure for Magnetoencephalography, Karolinska Institutet, Sweden (www.natmeg.se). Data were also in part provided by the Human Connectome Project, WU-Minn Consortium (Principal Investigators: David Van Essen and Kamil Ugurbil; 1U54MH091657) funded by the 16 NIH Institutes and Centers that support the NIH Blueprint for Neuroscience Research; and by the McDonnell Center for Systems Neuroscience at Washington University. Funding was provided to T.K. by Marianne and Marcus Wallenbergs Stiftelse (MMW 2020.0064).

## Author contributions

J.E. and T.K. conceived the study. J.E. and N.R.I. analyzed the data. J.E., T.K. and M.L. interpreted the results. J.E. and T.K. wrote the first draft of the manuscript. Revision was made by all authors. T.K. administrated the project.

## Funding

## Competing interests

The authors declare no competing interests.
