## [Transparent Peer Review file · Nature Communications]

Low frequency oscillations – neural correlates of stability and flexibility in cognition

Corresponding Author: Professor Torkel Klingberg

Version 0:

Reviewer comments:

Reviewer #1

(Remarks to the Author)

In this reported study MEG data are analysed, and it is demonstrated that early posterior theta synchronisation during intake of information seems to be associated with the encoding of information into working memory; whereas during maintenance of information in working memory dorsal alpha activity seems relevant for shielding this information from distraction. By using computational brain models it is also nicely demonstrated that these two different oscillatory brain states are associated with different patterns of information flow in the cortex.

In general, this manuscript has been a super exciting read. I really enjoyed it. The used approach (combination of MEG data analysis and modeling) is cutting edge, and is an approach that should be used so much more in cognitive neuroscience. It is absolutely beyond questioning that this manuscript is highly innovative. The addressed topic is exciting; and the paper is very well written and easy to follow (maybe some more details could be provided some times - but see below for specific comments).

Although I am fascinated by the approach and the manuscript, I think there are certain issues that should be addressed to make the results even more convincing.

1) Posterior theta synchronisation goes up during stimulus presentation, and central (sensorimotor) alpha synchronisation is increased when participants are not confronted with a new stimulus. The interpretation is that posterior theta synchronisation stands for encoding of new information, central alpha for protection against distraction. But could one not also conclude that these effects are largely independent of working memory? When a stimulus is presented one would expect a strong evoked response over visual areas. It is common that this evoked response resembles characteristics of theta waves (let us not even start discussing whether those apparent theta waves are true oscillations or are just an evoked response that look like theta). Therefore, one would get spurious theta synchrony right after stimulus onset. Independent of any true memory effect.

Between stimuli, in most of the trials, participants will have to withhold a response (e.g. non-target trials in the n-back task; not responding in the grid task). There is quite a bit of literature which suggests that withholding motor responses can lead to synchronisation in the alpha band over sensorimotor areas similar as in the current task. I think, to convincingly show that theta and alpha synchronisation actually represent the memory functions proposed in the manuscript, a control experiment would need to be run in which no working memory component what so ever is present. If the described posterior theta and central alpha dynamics are then absent, this would be quite convincing evidence for the interpretation suggested in the manuscript.

2) Assuming that the reported correlates are really working memory relevant: There is recent work by Kia Nobre, Freek van Ede and Sage Boettcher suggesting that action plans reflected by modulation in alpha activity in the sensorimotor cortex can support visual working memory traces. The alpha modulations reported e.g. in Boettcher et al. 2021 (Sci Adv) reminded me a lot of your stage 3. I can imagine it would be interesting linking these two streams of research.

3) I feel as there should be more detail on the results included in the manuscript. There are only a few p-values reported. It is necessary that proper statistical test values are included (also effect sizes). For the simulation it is sometimes claimed that certain effects are more significant than others, but again there is no statistics reported what so ever.

Minor:

- a) When describing the experimental paradigms any kind of stimulus is called "cue". Why not calling them stimuli instead?
- b) In the methods section 2.1 it should read "During the remaining days, ..."
- c) in the merged pdf the abstract was somehow messed up. Luckily the word source file was fine.

(Remarks on code availability)

Reviewer #2

(Remarks to the Author)

In the present manuscript, Ericson et al. analyze multiple MEG datasets during simple WM operations, which they combine with simulation approaches, to demonstrate that two cortical states (theta and alpha synchrony) support working memory operations in humans.

The analyses provide a unique angle on the WM literature and combine several complementary approaches to investigate the role of theta/alpha activity for WM.

While I am generally enthusiastic about the approach and the combination of multiple datasets/methods/approaches, several issues hamper the interpretations and make the interpretations a bit more difficult.

First and foremost, the authors start out by filtering their data into two canonical frequency bands, without establishing that these are actually characterized by distinct peaks in the power spectrum. Hence, this might conflate background and true oscillatory activity and explain why theta and alpha are partially overlapping. Second, it seems problematic to state that one identified two different modes (theta and alpha), when the data was already a-priori selected in those two ranges.

Major points:

- The paper in its current form lacks a coherent story. It seems that this can be partially attributed to a writing style for a condensed format. While it is fantastic that the authors carried out both experimental and computational work, the manuscript would benefit from a clearer description why subsequently described steps follow from each other.

- The title is very broad and very general and not well connected to the main paper. Given the lack of behavioral support and only indirect testing of flexible and stable cognition, the title seems a bit disconnected from the actual data reported

- The section "In-silico simulations of encoding and maintenance" is very methods-dense and hampers the flow of the paper.

- The manuscript oftentimes lacks appropriate statistical reporting. In several instances (i.e., line 408-410), no statistics are reported at all. If statistics are being reported, this is usually constrained to the p-value. I suggest adding the statistical test being used and more carefully describe what has been tested. For example, line 283-285: "This analysis revealed a significant similarity across tasks [...] (all $p < \dots$)". Which test has been used?

- Relatedly, it is oftentimes not clear which timepoints were used for specific analysis. This makes it difficult to assess the interpretation of the results.

- Figures:

1. Relatively small; consider increasing the font-size of the legends etc.

2. Figure 2, Figure 4c-d are missing a colorbar

3. The colorbars in Figure 5 & 6 are missing a label (i.e. what is being shown) and are very small

4. The figure legends only provide very little information (i.e. Figure 1 on task design). It would be easier for the reader if some essential information were included in the figure legends. Similarly, what does "mean response" in Figure 3B refer to. Please carefully check all figure legends and add information where appropriate.

- The tasks are explained and depicted (Figure 1) with only very little information which makes it very difficult to understand them. I would advise the authors to explain the experimental procedures in more detail instead of simply referring to a prior publication. For example, from the author's prior publication (Ericson 2024), I understand that participants were instructed to respond verbally during the WM grid task. Was this also the case in the current study? This would at least explain the numbers in Figure 1b. Please specify how participants were instructed to respond across all three tasks.

- The authors state that "the states were more closely related to task demands" (line 301). Where is the data that supports this interpretation? To support this interpretation, the authors could assess load-dependent differences within the states? For example, does the posterior alpha-synchronization increase as a function of working memory load? This could be assessed in the N-back task or in the WM-grid task where load is enhanced across time.

- The theta- and alpha-state-specific distributions are quite similar which is surprising given the prominent "frontal theta" reported in the WM literature. To further strengthen the point, that these two reflect two distinct states, it might be beneficial first establish the presence of distinct oscillations and then to re-run the cosine similarity analysis across distinct states (i.e., compute the similarity between posterior theta and posterior alpha across states etc.) to show that the cosine similarity across similar states is higher as compared to the cosine similarity across different states.

- I am a bit surprised that the authors only computed the information flow by means of transfer entropy on the simulated data. I think it would make a stronger case if they could replicate this using the experimental data at hand. The authors could for example use the parcels strongly contributing to a specific state and compute the transfer entropy during both the cue and maintenance period to all other parcels. Maybe the authors have a good reason as to why they did not perform the analyses on the actual data, but it should be at least addressed in the manuscript.

- How do the authors relate their findings to prior studies that showed an increase in fronto-parietal theta synchrony specific to the maintenance period (i.e. Jacobs et al. Neuron 2018)? The reason I am asking is because I am a bit concerned that the early rise in theta synchronization upon the cue (cf. Figure 3c, state 1) might be strongly driven by the ERP. Could the authors show that this is really specific to the theta frequency band? One way would be to compute the theta power on the ERP trace and subtract it from power traces upon which ICA is being performed.

- Similarly, could the authors plot some raw power traces in the theta/alpha frequency range? This would help to better understand the overall frequency-specific dynamics in the task. Moreover, I would suggest that the authors visualize some spectra to show that the states are actually driven by "true oscillations" and not by something rather aperiodic/background activity in nature.

- Overall, the paper's claims would benefit strongly from some analyses linking their findings to behavior. For example, the author's state that "stronger dorsal alpha synchronization leads to better maintenance..." (line 474). Do the authors have some data that could substantiate this interpretation? It would be great if the authors could link the strength of network synchronization to behavior. This would also help to link their findings to stability and flexibility in cognition, as suggested in the title.

Minor:

- I would suggest that the authors use a better visualization for the table

- Figure 3b: "The y-axis represents the portion of subjects classified into each state at each timepoint". This does not really fit with the actual y-label ("portion of time spent in state")

- The authors state that the time-frequency compromise parameter was set to 5. I assume that they mean the number of cycles per wavelet. I would suggest keeping the more technical nomenclature to avoid confusion.

Taken together, while the topic is timely and interesting, the manuscript requires substantial additional analyses to strengthen its conclusions. That being said, I am positive that all concerns can be addressed in a revision.

(Remarks on code availability)

I could not open the second code file, the first one is rather sparse, but seems like a straight-forward implementation

Reviewer #3

(Remarks to the Author)

See attachment.

(Remarks on code availability)

Version 1:

Reviewer comments:

Reviewer #1

(Remarks to the Author)

The authors have put a lot of effort into the revision. This is very much appreciated. All my concerns have been addressed very adequately. I find the additional evidence (the control data) very convincing. I want to thank the authors for taking the raised issues so serious.

The revised manuscript really is a great read. As already pointed out in the first reviewing round, this is a brilliant and innovative approach followed through in the study. I think the paper will be very well received by the scientific community.

(Remarks on code availability)

Reviewer #2

(Remarks to the Author)

The authors addressed all queries in detail. I also found the comments and responses to the queries raised by the other reviewers very helpful. The last query remains based on the new data, how do the authors explain the inverted u-shaped curve for WM load-dependence and time spent in a specific state? How can this be reconciled with previous WM theories (shortening or lengthening of the theta cycle?, e.g., the Lisman model)? Could it be that the apparent decrease of time spent in state 3 stems from a peak frequency shift, which might cause that the component is no longer identified as such?

Other than that, the authors were remarkably responsive and provided an in-depth response that substantially improved the manuscript.

(Remarks on code availability)

No additional issues.

Dear Editor,

Thank you for considering our manuscript. We have now substantially revised it according to all of the reviewers' concerns. The new manuscript includes both new analyses and two new MEG datasets to support our claims.

First, we have included a new analysis of a control task without WM requirements, performed by the same subjects included in the previous version of the manuscript. We could show that the systematic switch between the encoding and maintenance states disappear in the control task. Second, we show that the time spent in the two states are dependent on WM load, something both reviewers asked about. Thirdly, we analyzed a new dataset acquired during WM performance with and without distractors from a new set of subjects. We show that the time spent in the encoding state was regulated differently for distractor trials compared to no distractor trials even though the stimuli were identical.

We have also added an analysis of a new behavioral dataset from the Human Connectome Project to show that the control of state-switching is associated with cognitive ability. Finally, we expanded on the simulations to also investigate the impact of different synchronization frequencies on information flow.

In all, the new data confirms our prior conclusions, and the additional simulations and behavioral correlations increase the novelty of the results.

Source data to all figures and table is now provided in the folder SourceData. Apart from the data, it includes a notebook detailing how the data is visualized. Regarding the formatting requirements, we have a question. We understood from your previous email that bar graphs should be avoided. However, for better interpretability and since the zero mark has a physical meaning (no information transfer) is it possible to keep the bar graphs if we also include the individual data points (see below)?

Below please find our point-by-point answers.

Best regards,

Julia Ericson, Nieves Ruiz Ibáñez, Mikael Lundqvist, and Torkel Klinberg

REVIEWER COMMENTS

Reviewer #1 (Remarks to the Author):

In this reported study MEG data are analysed, and it is demonstrated that early posterior theta synchronisation during intake of information seems to be associated with the encoding of information into working memory; whereas during maintenance of information in working memory dorsal alpha activity seems relevant for shielding this information from distraction. By using computational brain models it is also nicely demonstrated that these two different oscillatory brain states are associated with different patterns of information flow in the cortex.

In general, this manuscript has been a super exciting read. I really enjoyed it. The used approach (combination of MEG data analysis and modelling) is cutting edge and is an approach that should be used so much more in cognitive neuroscience. It is absolutely beyond questioning that this manuscript is highly innovative. The addressed topic is exciting; and the paper is very well written and easy to follow (maybe some more details could be provided sometimes - but see below for specific comments).

Although I am fascinated by the approach and the manuscript, I think there are certain issues that should be addressed to make the results even more convincing.

1) Posterior theta synchronisation goes up during stimulus presentation, and central (sensorimotor) alpha synchronisation is increased when participants are not confronted with a new stimulus. The interpretation is that posterior theta synchronisation stands for encoding of new information, central alpha for protection against distraction. But could one not also conclude that these effects are largely independent of working memory? When a stimulus is presented one would expect a strong evoked response over visual areas. It is common that this evoked response resembles characteristics of theta waves (let us not even start discussing whether those apparent theta waves are true oscillations or are just an evoked response that look like theta). Therefore, one would get spurious theta synchrony right after stimulus onset. Independent of any true memory effect. Between stimuli, in most of the trials, participants will have to withhold a response (e.g. non-target trials in the n-back task; not responding in the grid task). There is quite a bit of literature which suggests that withholding motor responses can lead to synchronisation in the alpha band over sensorimotor areas similar as in the current task. I think, to convincingly show that theta and alpha synchronisation actually represent the memory functions proposed in the manuscript, a control experiment would need to be run in which no working memory component what so ever is present. If the described posterior theta and central alpha dynamics are then absent, this would be quite convincing evidence for the interpretation suggested in the manuscript.

Response:

Thank you for reviewing the article, we are glad you liked it.

You ask for a control task to show that dorsal alpha and posterior theta actually represent memory functions.

To show that theta is related to encoding, we have included a new dataset (n=13) of a vsWM task with and without distractors. In the distractor trials, stimuli 2 and 3 were not to be encoded. If the theta synchronisation was not related to task demands, but only to visual stimulation, identical stimuli would give identical theta responses independent of whether the stimuli are distractors or not. In the new analysis, we show that, on the contrary, participants decreased the time spent in the posterior theta state for distractors (see new Figure 7). Moreover, if the trial included distractors, participants spent more time in the posterior theta state during stimulus 4 (which for distractor trials is the second load instead of the fourth). This is also in line with our new load dependency analysis (new Figure 6), showing a decrease of posterior theta as cognitive load increases.

time spent in the posterior theta state for distractor trials compared to no distractor trials. Before the trial starts, a cue was shown to indicate if the trial was a distractor or a no distractor trial. At 0.5, 1.5, 2.5, and 3.5 seconds, a stimulus was shown for 0.5 seconds, followed by a 0.5 second delay period. The shaded areas mark the presentation periods. The subject was to remember the rotation of the bar and the color of the dot in the middle of the bar. After presentation of the four items, one of the four colors were shown, telling the subject to recall the

The theta component did not disappear completely for distractors; however, as shown here, the task relevant stimuli elicit stronger responses.

To remove both the alpha and the theta components completely as you asked for, we included an analysis of a control task which was part of the 4-subject dataset. The control task was a verbal recognition task with the same timings as the vsWM tasks (1 s delay periods between presentation of 5 to 6 items, after which the subjects used a button to respond). Here, we found that the encoding/maintenance pattern disappeared when the states were not time-locked to any specific task demands (new Fig. 6a and b):

rior theta state

In all, these new analyses strengthen the association between encoding and posterior theta, and between maintenance and dorsal alpha.

Further, you ask about the possibility that the theta synchronization could be a spurious oscillation in the evoked response. The theta oscillation is indeed phase-locked to the onset of the stimulus, but this is not mutually exclusive to the component being used to control information flow. For example, in mice it has been shown that theta waves, phase-locked to the onset of visual stimuli, synchronize high-frequency activity between visual and parietal cortex through phase amplitude coupling (Aggarwal et al 2022). Moreover, the modulation of the theta component according to task demands is not something that would be expected if the component was independent of any memory effect.

Finally, you also ask about the possible association between dorsal alpha and withholding a motor response. We don't think this can be the case because for the WM-Grid task, the subjects responded verbally, and for the 2-back task, participants reported their answer for both target and non-target trials. In the 2-back task the rise in dorsal alpha also occurs more than 0.5 seconds after the average response time.

The only periods which could have motor inhibition are the delays during Odd One Out and the control task. In Odd One Out, the participants spend less time in the dorsal alpha state compared to WM-Grid, and in the control task, there is no rise in dorsal alpha during the delay periods. Therefore, it is unlikely that the systematic rise in dorsal alpha during delay periods is related to motor inhibition. We have added further descriptions of how the participants respond to each task under the results section 'Data'.

Having said that, we also want to point out that these brain states could represent stability vs flexibility also in other visuospatial tasks, and we are not claiming that they are exclusive to vsWM. For example, we would assume that dorsal alpha would be involved whenever visual and/or spatial stability is needed (e.g. reading, visual attention, eye-movements, etc.), perhaps explaining why similar patterns could be present during motor inhibition. Likewise, posterior theta would appear whenever visual flexibility and sampling of new visual information is needed.

Thus, we interpret the states to represent fundamental cognitive processes, which could be manipulated during cognitive tasks to achieve the desired goal (for example, decreasing posterior theta during distractors presentation).

We have made updates throughout the introduction and discussion to convey the message about state generality better. For example, in the first paragraph of the introduction we write: "Even in simple tasks such as visual perception, the brain needs to alternate between flexibly sampling new information and generating visual stability during eye movements".

2) Assuming that the reported correlates are really working memory relevant: There is recent work by Kia Nobre, Freek van Ede and Sage Boettcher suggesting that

action plans reflected by modulation in alpha activity in the sensorimotor cortex can support visual working memory traces. The alpha modulations reported e.g. in Boettcher et al. 2021 (Sci Adv) reminded me a lot of your stage 3. I can imagine it would be interesting linking these two streams of research.

Response:

Thank you for these suggestions. We were not aware of this work, but it is in line with how we reason about these networks. The results from Boettcher et al are now referenced in paragraph three of the discussion, where we review dorsal alpha during other tasks requiring spatial stability.

3) I feel as there should be more detail on the results included in the manuscript. There are only a few p-values reported. It is necessary that proper statistical test values are included (also effect sizes). For the simulation it is sometimes claimed that certain effects are more significant than others, but again there is no statistics reported what so ever.

Response:

We have now added p-values, effect sizes, t-values and confidence intervals for all comparisons.

Minor:

a) When describing the experimental paradigms any kind of stimulus is called "cue". Why not calling them stimuli instead?

Response: *Yes, they should be called stimuli. We have now changed that throughout the manuscript.*

b) In the methods section 2.1 it should read "During the remaining days, ..."

Response: *Thank you for noticing this, it is now changed.*

c) in the merged pdf the abstract was somehow messed up. Luckily the word source file was fine.

Response: *Thanks, we will have an extra look at that for the next submission.*

Reviewer #2 (Remarks to the Author):

In the present manuscript, Ericson et al. analyze multiple MEG datasets during simple WM operations, which they combine with simulation approaches, to demonstrate that two cortical states (theta and alpha synchrony) support working memory operations in humans. The analyses provide a unique angle on the WM literature and combine several complementary approaches to investigate the role of theta/alpha activity for WM.

While I am generally enthusiastic about the approach and the combination of multiple datasets/methods/approaches, several issues hamper the interpretations and make the interpretations a bit more difficult.

- First and foremost, the authors start out by filtering their data into two canonical frequency bands, without establishing that these are actually characterized by distinct peaks in the power spectrum. Hence, this might conflate background and true oscillatory activity and explain why theta and alpha are partially overlapping.

Response:

We are happy you enjoyed the manuscript and hope that the following alterations will make the interpretations of the results easier.

Most importantly, we have now added Figure 2, which shows a spectrum of synchronization within lower frequencies. In Figure 2, we also plotted the temporal dynamics of the two frequency bands before the ICA is applied, to show the underlying dynamics within the frequency bands for both datasets. We used synchronization instead of power as it generally removes more of the aperiodic activity.

Second, it seems problematic to state that one identified two different modes (theta and alpha), when the data was already a-priori selected in those two ranges.

Response:

We did not identify one alpha and one theta mode. In fact, the independent component analysis showed that there were two independent signals within each frequency band, rendering four states in total.

We believe the large focus we put on two of these states (the posterior theta and the dorsal alpha) might have led to the misconception that there is only one alpha state and one theta state. Therefore, in the updated version of the manuscript, we have put more emphasis on the other two states as well, both through additional behavioral analysis of the HCP dataset (see section “State-switches and cognitive performance” under Results) and through additional simulations (see section “The influence of frequency on information flow” under Results).

Finally, while it may not be surprising to find different modes in different frequencies using our approach, it is not obvious that they should consistently correlate with distinct cognitive behaviors.

Major points:

- The paper in its current form lacks a coherent story. It seems that this can be partially attributed to a writing style for a condensed format. While it is fantastic that the authors carried out both experimental and computational work, the manuscript would benefit from a clearer description why subsequently described steps follow from each other.

Response:

We agree with this remark and have revised Abstract, Introduction, Results and Discussion with this concern in mind. All changes are highlighted. We believe the manuscript is now more coherent and follows a more visible story line.

- The title is very broad and very general and not well connected to the main paper. Given the lack of behavioral support and only indirect testing of flexible and stable cognition, the title seems a bit disconnected from the actual data reported

Response:

We have updated the title to “Low frequency oscillations – neural correlates of stability and flexibility in cognition”. Furthermore, in the new version of the manuscript, we include several new analyses which link our results to cognitive behavior more broadly:

- 1) *For the HCP dataset, we have included an analysis of the relationship between state-switching and measures of cognitive flexibility, inhibitory control and processing speed.*
- 2) *For the 4-subject dataset, we added a control task. The task had the same timings as the vsWM tasks, but instead of vsWM it was a verbal recognition task. Here, the systematic switching between states 1 and 3 during stimulus presentation and delay disappeared.*
- 3) *For the 4-subject dataset, we also added a study of load dependency to further relate the states to vsWM.*
- 4) *We added a new dataset with distractors to show that encoding/theta state duration differed depending on whether the stimulus was a distractor or not.*

We believe that this has strengthened the connection between the neural pattern we have observed and flexible and stable cognitive states.

- The section "In-silico simulations of encoding and maintenance" is very methods-dense and hampers the flow of the paper.

Response:

Thank you for pointing this out. We have moved as much as possible of the methods describing the simulations (including all equations) to the method section.

- The manuscript oftentimes lacks appropriate statistical reporting. In several instances (i.e., line 408-410), no statistics are reported at all. If statistics are being reported, this is usually constrained to the p-value. I suggest adding the statistical test being used and more carefully describe what has been tested. For example, line 283-285: "This analysis revealed a significant similarity across tasks [...] (all $p < \dots$)". Which test has been used?

Response:

We have now added p-values, effect sizes, t-values and confidence intervals for all comparisons.

- Relatedly, it is oftentimes not clear which timepoints were used for specific analysis. This makes it difficult to assess the interpretation of the results.

Response:

We have read through all analyses to make sure that the timepoints used for each analysis are clearly stated. We have specifically clarified the periods from which

filtered data were extracted under the Results section “Identification of Networks”. We hope this clarifies the interpretation of the results.

- Figures:

1. Relatively small; consider increasing the font-size of the legends etc.
 2. Figure 2, Figure 4c-d are missing a colorbar
 3. The colorbars in Figure 5 & 6 are missing a label (i.e. what is being shown) and are very small
 4. The figure legends only provide very little information (i.e. Figure 1 on task design). It would be easier for the reader if some essential information were included in the figure legends. Similarly, what does “mean response” in Figure 3B refer to. Please carefully check all figure legends and add information where appropriate.
-

Response:

Thank you for these suggestions. We have made the font size larger, added color bars and color bar legends where that was missing, and expanded on the figure legends. “Mean response” is the mean time it takes for participants to respond. We have changed the wording here to “mean response time” to make it clearer.

- The tasks are explained and depicted (Figure 1) with only very little information which makes it very difficult to understand them. I would advise the authors to explain the experimental procedures in more detail instead of simply referring to a prior publication. For example, from the author’s prior publication (Ericson 2024), I understand that participants were instructed to respond verbally during the WM grid task. Was this also the case in the current study? This would at least explain the numbers in Figure 1b. Please specify how participants were instructed to respond across all three tasks.

Response:

Yes, that was also the case in the current study. We have added more details for all tasks, including how participants were instructed to respond under the results section ‘Data’.

- The authors state that “the states were more closely related to task demands” (line 301). Where is the data that supports this interpretation? To support this interpretation, the authors could assess load-dependent differences within the states? For example, does the posterior alpha-synchronization increase as a function of working memory load? This could be assessed in the N-back task or in the WM-grid task where load is enhanced across time.

Response:

As suggested, we have now included a new analysis of load dependency for the WM-Grid and the Odd One Out task, both for dorsal alpha and posterior theta. The analysis shows that posterior theta decreased as a function of load. The same pattern also held for a new dataset with distractor stimuli (see section “The control of posterior theta” under Results), where the fourth stimulus had WM load 4 in the non-distractor trials but WM load 2 in distractor trials.

Dorsal alpha initially increased as a function of load and then decreased. The decrease in alpha occurs after load 3, aligned with the WM load where recall errors typically start to occur. There was also a trend where those subjects with higher accuracy also had less decrease in alpha after load 3. However, with only 4 subjects we can't draw any conclusions from this. The results are presented in the new figure 6 (d and e):

m, and load 5 is the 1000 ms delay period after the presentation of the fifth item. Below is the time spent in state 1 during the 300 ms presentation of each stimulus 1 – 5. e) The time spent in states 1 and 3 during the Odd One

Finally, we agree that the sentence “the states were more closely related to task demands” was unclear. We have now removed it.

- The theta- and alpha-state-specific distributions are quite similar which is surprising given the prominent “frontal theta” reported in the WM literature. To further strengthen the point, that these two reflect two distinct states, it might be beneficial first establish the presence of distinct oscillations and then to re-run the cosine similarity analysis across distinct states (i.e., compute the similarity between

posterior theta and posterior alpha across states etc.) to show that the cosine similarity across similar states is higher as compared to the cosine similarity across different states.

Response:

The additional synchronization analysis in Figure 2 establishes that there are distinct oscillations in the alpha and theta bands, which have different temporal patterns. Spatially, the networks are very similar (cosine similarity around 0.6); however, the dominating frequency changes over time, as identified by the clustering analysis resulting in the states. Thus, the states are defined by both a frequency and a location.

However, it is interesting that the synchronization frequency within similar areas systematically changes for different time periods of the tasks. This suggests that, in addition to the location of the synchronization being important, the frequency itself is also important. We have explored this further in a new simulation where we show that the frequency of the oscillations influences how information is routed (new figure 9d). This further illustrated why it is important to take both frequency and location into account when defining cognitive states.

- I am a bit surprised that the authors only computed the information flow by means of transfer entropy on the simulated data. I think it would make a stronger case if they could replicate this using the experimental data at hand. The authors could for example use the parcels strongly contributing to a specific state and compute the transfer entropy during both the cue and maintenance period to all other parcels. Maybe the authors have a good reason as to why they did not perform the analyses on the actual data, but it should be at least addressed in the manuscript.

Response:

The reason for not including experimental data here has mainly to do with the fact that the amount of data is not enough to get a good measure of transfer entropy. Even though there are many advantages with Transfer Entropy, such as the ability to analyze non-linear causality, it also requires much more continuous data than we had. This refers specifically to the numerical estimations of the probability distributions. For example, during simulations we used 100,000 timepoints to get a stable estimate of transfer entropy between two regions.

Furthermore, desynchronized activity from higher frequency bands is difficult to measure using MEG where the signal to noise ratio is lower for high compared to low frequencies. We have added an explanation for this in the results section under 'In-silico Simulations of Encoding and Maintenance'.

- How do the authors relate their findings to prior studies that showed an increase in fronto-parietal theta synchrony specific to the maintenance period (i.e. Jacobs et al. Neuron 2018)? The reason I am asking is because I am a bit concerned that the early rise in theta synchronization upon the cue (cf. Figure 3c, state 1) might be strongly driven by the ERP. Could the authors show that this is really specific to the theta frequency band? One way would be to compute the theta power on the ERP trace and subtract it from power traces upon which ICA is being performed.

Response:

If we look at the raw data from the regions that are part of the posterior network, we see a distinct theta oscillation upon cue (see the left figure below). However, these oscillations are time-locked to the onset of the stimulus and therefore, they will also show up in the ERP trace (see the right figure below). The fact that the component is phase-locked to the onset of the stimulus does not necessary mean that it cannot be used as the control mechanism suggested in this manuscript. For example, in mice it has been shown that theta waves, phase-locked to the onset of visual stimuli, coordinate high-frequency activity between visual and parietal cortex through phase amplitude coupling (Aggarwal et al 2022).

We could further demonstrate the active involvement of the theta component in the tasks by showing how it was modulated differently for different task demands. More

specifically, we added a new analysis with distractors, where we could show that the posterior theta state was downregulated during distractors.

Secondly, in the analysis of load dependency in the 4-subject dataset, we also found that theta was downregulated for higher loads. The same pattern was seen in the distractor dataset where the fourth stimulus could either have cognitive load 2 or 4 depending on whether the trial has distractors or not (new Fig. 7):

time spent in the posterior theta state for distractor trials compared to no distractor trials. Before the trial starts, a cue was shown to indicate if the trial was a distractor or a no distractor trial. At 0.5, 1.5, 2.5, and 3.5 seconds, a stimulus was shown for 0.5 seconds, followed by a 0.5 second delay period. The shaded areas mark the presentation of the stimuli. The subject was to remember the rotation of the bar and the color of the dot in the middle of the bar. After presentation of the four items, one of the four colors were shown, telling the subject to recall the rotation of the matching bar.

Finally, we have also elaborated on the absence of a frontal theta network in the discussion with a reference to Jacob et al. Many vsWM studies in humans lack a frontal theta component, even though it is often observed in monkeys. It is possible that this can be explained by the extensive training necessary for monkeys to complete the task. As mentioned in the manuscript, we also found the emergence of a frontal midline theta component over time in the 4-subject dataset.

- Similarly, could the authors plot some raw power traces in the theta/alpha frequency range? This would help to better understand the overall frequency-specific dynamics in the task. Moreover, I would suggest that the authors visualize some spectra to show that the states are actually driven by “true oscillations” and not by something rather aperiodic/background activity in nature.

Response:

As requested, we have now added a new figure (Fig 2) which shows a synchronization spectrum and raw synchronization traces:

We used synchronization instead of power because synchronization removes more of the aperiodic activity compared to power.

- Overall, the paper's claims would benefit strongly from some analyses linking their findings to behavior. For example, the author's state that "stronger dorsal alpha synchronization leads to better maintenance..." (line 474). Do the authors have some data that could substantiate this interpretation? It would be great if the authors could link the strength of network synchronization to behavior. This would also help to link their findings to stability and flexibility in cognition, as suggested in the title.

Response:

We have performed an additional behavioral analysis of the HCP data. This demonstrated a significant correlation between the control of state-switches to cognitive performance (new Figure 5) as shown below. More specifically, we found an optimal state-switching rate where fewer or more frequent switches were associated with lower cognitive performance:

nce. a) The correlation between the tasks in the test battery: Flanker's Inhibitory Control and Attention Task,

We generally focused on measures that could relate the states to behavior (i.e. state switches and state duration) instead of synchronization strength itself. This was because the focus of the article was on states, and we wanted to limit the number of relationships we tested. However, as mentioned in the discussion, there are previous studies which demonstrate the link between alpha synchronization in dorsal areas

and vsWM capacity, as well as between posterior theta strength and memory encoding.

Minor:

- I would suggest that the authors use a better visualization for the table

Response: *From what we understand from nature communications “Tables must be provided in an editable format and prepared using the table menu in Word or the table environment in LaTeX.”*

Or have we misunderstood your concern? Do you think we should use a figure such as a box plot instead of a table?

- Figure 3b: “The y-axis represents the portion of subjects classified into each state at each timepoint”. This does not really fit with the actual y-label (“portion of time spent in state”)

Response: *Thank you for pointing this out. We have now changed the label to “Portion of subjects in state”.*

- The authors state that the time-frequency compromise parameter was set to 5. I assume that they mean the number of cycles per wavelet. I would suggest keeping the more technical nomenclature to avoid confusion.

Response: *Correct, we have changed this to “number of cycles per wavelet”.*

Taken together, while the topic is timely and interesting, the manuscript requires substantial additional analyses to strengthen its conclusions. That being said, I am positive that all concerns can be addressed in a revision.

Response: *Thank you for the constructive comments. We indeed believe that the requested analysis substantiated our claims, and that text edits improved the interpretability of the results by including more intermediate steps.*

Reviewer #2 (Remarks on code availability):

I could not open the second code file, the first one is rather sparse, but seems like a straight-forward implementation

Response: *The second file is a jupyter notebook and can't be opened as a text file. Could that be why you had issues? The code is also available on GitHub now (link under Code availability in the manuscript) and there, notebooks can be read (in case you do not have a program installed which can read notebooks). Hopefully this will solve the problem, please let us know otherwise!*

Reviewer #3 (Remarks to the Author):

I co-reviewed this manuscript with one of the reviewers who provided the listed

reports. This is part of the Nature Communications initiative to facilitate training in peer review and to provide appropriate recognition for Early Career Researchers who co-review manuscripts.

Response: *Thank you for taking the time to co-review the manuscript.*

Dear editor,

We have now revised the manuscript according to the editorial requests and the remaining comments from reviewer 2. We evaluated the possibility that the peak frequency could shift as a function of load, as suggested by reviewer 2. However, we did not find such a shift. A figure from this analysis has been included in the supplementary.

In addition, the following changes have been made:

- We included a summary figure in the supplementary as requested in the Author Checklist.
- We added two new supplementary figures that were previously included in the Response to Reviewers (supplementary Fig. 1 and 3).
- We added or made updates to the sections Code Availability, Acknowledgements, Author contribution, and Competing Interests as requested in the Author Checklist.

Finally, here is a suggested editor's summary:

How does the brain balance the flexibility and stability needed to both encode and maintain information during cognitive tasks? Using MEG data and in-silico simulations, the authors show that neural oscillations can be used as a dynamic control mechanism to shift between flexible and stable brain states.

A point-by-point response to the reviewers' comments is included below.

Sincerely, and on behalf of all the co-authors,
Torkel Klingberg and Julia Ericson

REVIEWERS' COMMENTS

Reviewer #1 (Remarks to the Author):

The authors have put a lot of effort into the revision. This is very much appreciated. All my concerns have been addressed very adequately. I find the additional evidence (the control data) very convincing. I want to thank the authors for taking the raised issues so serious.

The revised manuscript really is a great read. As already pointed out in the first reviewing round, this is a brilliant and innovative approach followed through in the study. I think the paper will be very well received by the scientific community.

Reviewer #2 (Remarks to the Author):

The authors addressed all queries in detail. I also found the comments and responses to the queries raised by the other reviewers very helpful. The last query the remains based on the new data, how do the authors explain the inverted u-shaped curve for WM

load-dependence and time spent in a specific state? How can this be reconciled with previous WM theories (shortening or lengthening of the theta cycle?, e.g., the Lisman model)? Could it be that the apparent decrease of time spent in state 3 stems from a peak frequency shift, which might cause that the component is no longer identified as such?

Other than that, the authors were remarkably responsive and provided an in-depth response that substantially improved the manuscript.

Response: We are happy to hear that the responses and comment were helpful. Regarding the inverted u-shape for state 3, we have evaluated the possibility of a frequency shift as suggested by the reviewer.

When we analyze the synchronization spectra for each load independently (delay period only), we see that the alpha synchronization peak remains stable (new supplementary figure 3). Therefore, we do not believe that a peak shift could explain the u-shaped curve.

Supplementary 3: The global synchronization for WM loads 0–5 averaged over participants and sessions, for a) WM-Grid and b) Odd One Out.

Another possibility is that the alpha network is already at maximum strength after a WM load of around 3 items (as is also suggested by the synchronization spectra above). This might lead to a change of strategy for higher loads, where subjects switch to rely more on other mechanisms, resulting in a state switch. We touch on this possibility in the discussion (page 10, paragraph 3):

“Additionally, the highest dorsal alpha signal occurred at a cognitive load of three, consistent with the number of items humans can hold in memory without making noticeable errors (Luck and Vogel 1997). Beyond this load, additional processes such as compression – relying on patterns in the data (Dehaene et al. 2022) – may supplement visuospatial stability in working memory tasks.”

Reviewer #2 (Remarks on code availability):

No additional issues.

Ericson et al seek out to understand how brain states participate during encoding and maintenance in working memory. The motivation and proposed approach of the work is timely and relevant for a mechanistic understanding of how brain states underlie updating and maintenance. However, there are major concerns regarding the framework and methods in this work. Please find them below.

1. The authors filtered the MEG data using Morlet wavelets at 6 and 11 Hz for theta and alpha oscillatory activity. However, there is no clear justification on why they used those single frequencies rather than the whole frequency band. If the reasoning is that frequency bands and not single frequencies characterize brain states, then it would be necessary to show that these findings generalize to signals filtered between ~4-8 Hz for theta and ~9-14 Hz for alpha.
2. Using ICA, the authors identify four networks: a posterior alpha network, a dorsal alpha network, a posterior theta network and a dorsal theta network. In the methods section, the authors mentioned that for each frequency *“the appropriate number of components used in the algorithm differed between participant and frequency but were always between two and five components”*. What is the variance of the data for which these four networks account? Were these always the top components? For purposes of replicability, it would be important to include this information.
3. The spiking layer in the model is influenced by the oscillatory layer through phase-amplitude coupling. The spiking activity is enhanced during the troughs and suppressed during the peaks of synchronized oscillations regardless of the frequency. However, there exists contradictory evidence for the case of theta. In fact, another body of evidence argues that peaks of theta represent windows of excitability and enhancement of sensory processing (<https://doi.org/10.1016/j.cobeha.2024.101433>, <https://doi.org/10.1016/j.tics.2018.11.009>). One compelling way to justify the design of the model could be demonstrating in the experimental data that gamma activity is coupled to troughs of alpha and theta during the occurrence of both states 1 and 3, respectively.
4. Ericson et al show that posterior theta synchronization states facilitated an increase in flow of information from V1 to the rest of the brain, whereas dorsal alpha states increased flow of information from the IPS. The authors also evaluated the flow of information from regions outside the networks and found that flow of information into the dorsal network was facilitated by alpha synchronization.
 - a. The authors argue that there is a flow of information of higher order association areas to the frontoparietal network. However, this statement begs for a more rigorous examination. Given that the authors are making use of the Schaefer atlas, they should (1) show a breakdown of how many of the regions from the networks identified by ICA belong to the frontoparietal network (and might as well other canonical networks as a negative control) (2) show entropy transfer to regions from the frontoparietal network and not only the dorsal and posterior networks. It could also provide insight to quantify the contribution of other networks to the transfer of information by grouping regions based on functional networks.
 - b. It would be important to include as a negative control the stimulation of a region that is not related to the task and compare against the transfer entropy values of V1 and IPS. Additionally, the manuscript would benefit from a more detailed reasoning behind why they used these two regions.

Minor comments:

1. Increase font size in all figures.
2. Include colorbar labels in figure 5 and 6